# Envisioning Beyond the Pixels: Benchmarking Reasoning-Informed Visual Editing

**Xiangyu Zhao**[1,2*], **Peiyuan Zhang**[3*], **Kexian Tang**[2,4*], **Xiaorong Zhu**[1*],
**Hao Li**[2], **Wenhao Chai**[5], **Zicheng Zhang**[1,2], **Renqiu Xia**[1,2],
**Guangtao Zhai**[1,2], **Junchi Yan**[1], **Hua Yang**[1✉], **Xue Yang**[1✉†], **Haodong Duan**[2✉†]

[1] ICISEE & SAIS & SAI, Shanghai Jiao Tong University    [2] Shanghai AI Laboratory
[3] Wuhan University    [4] Tsinghua University    [5] Princeton University
*\* Equal contribution*    ✉ *Corresponding author*    † *Project lead*

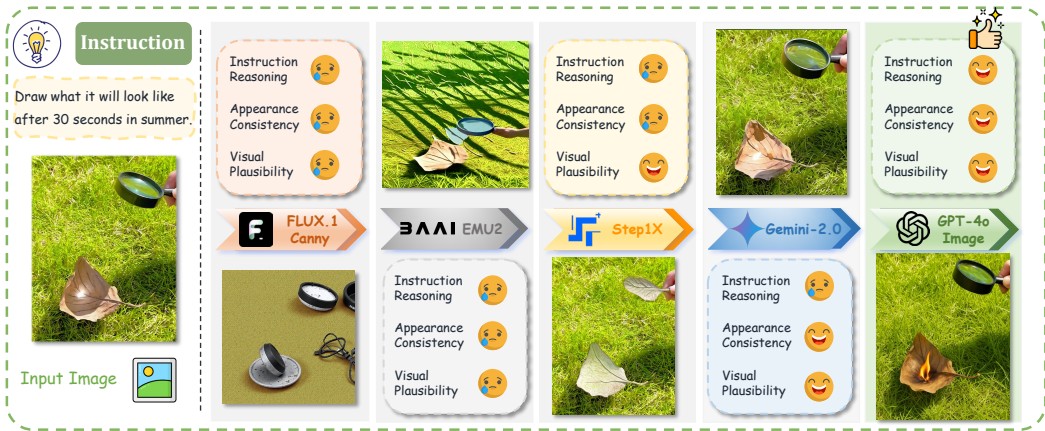

Figure 1: **Examples of leading models on the Reasoning-Informed viSual Editing(RISE) benchmark.** RISEBench contains complex and various tasks that pose challenges to current models.

## Abstract

Large Multi-modality Models (LMMs) have made significant progress in visual understanding and generation, but they still face challenges in General Visual Editing, particularly in following complex instructions, preserving appearance consistency, and supporting flexible input formats. To study this gap, we introduce **RISEBench**, the first benchmark for evaluating **R**easoning-**I**nformed vi**S**ual **E**diting (**RISE**). RISEBench focuses on four key reasoning categories: *Temporal*, *Causal*, *Spatial*, and *Logical Reasoning*. We curate high-quality test cases for each category and propose an robust evaluation framework that assesses *Instruction Reasoning*, *Appearance Consistency*, and *Visual Plausibility* with both human judges and the LMM-as-a-judge approach. We conducted experiments evaluating nine prominent visual editing models, comprising both open-source and proprietary models. The evaluation results demonstrate that current models face significant challenges in reasoning-based editing tasks. Even the most powerful model evaluated, GPT-image-1, achieves an accuracy of merely 28.8%. RISEBench effectively highlights the limitations of contemporary editing models, provides valuable insights, and indicates potential future directions for the field of reasoning-aware visual editing. Our code and data have been released at https://github.com/PhoenixZ810/RISEBench.

# 1 Introduction

Large Multi-Modality Models (LMMs) have achieved remarkable progress in both visual understanding [22, 1, 5, 38, 26] and visual generation [29, 31, 3]. Meanwhile, significant efforts [37, 48, 40, 42, 20] have been dedicated to unifying these two tasks, with the goal of enhancing overall performance through joint learning. Some open-source models have demonstrated decent capability in either visual understanding or image generation; however, they still exhibit substantial limitations in General Visual Editing (*i.e.*, transforming an input image based on textual instructions). Specifically, current open-source methods struggle with: (1) accurately following complex editing instructions [34]; (2) preserving the original image's appearance during visual editing [18]; and (3) accommodating flexible input formats [40, 48] (*e.g.*, supporting both single and multiple images with natural language instructions). These limitations severely hinder their practical utility, making them hardly worth rigorous evaluation in this task.

Recently, we observed that proprietary models such as GPT-image-1 [16] and Gemini-2.0-Flash* [38] have made significant advancements over open-source counterparts (Fig. 1). Notably, these models exhibit a remarkable capability in **R**easoning-**I**nformed vi**S**ual **E**diting (**RISE**) – a sophisticated ability that enables models to make intelligent visual modifications based on contextual understanding and logical reasoning. This advanced ability has exciting implications for various real-world applications, such as context-aware image modification (*e.g.*, adjusting lighting to match a scene's time of day), intelligent object insertion or removal with semantic consistency, and content adaptation based on inferred user intent. However, traditional image editing models [17, 11, 4] that do not incorporate multi-modal reasoning lack these capabilities entirely. While such a phenomenon is promising, we found that there is no well-established benchmark for systematically evaluating RISE task, making it difficult to quantitatively assess and further study this ability in existing models.

To this end, we introduce **RISEBench**, a focused, small-scale benchmark specifically designed to evaluate reasoning-informed visual editing (RISE) capabilities. In this benchmark, we identify and categorize key image editing challenges that require four fundamental types of reasoning: **temporal reasoning**, **causal reasoning**, **spatial reasoning**, and **logical reasoning**. To ensure a comprehensive evaluation, we manually curated a diverse set of high-quality test cases across the four categories: 85 for temporal reasoning, 90 for causal reasoning, 100 for spatial reasoning, and 85 for logical reasoning, resulting in a total of **360 carefully human-annotated samples**.

For evaluation, we decompose the quality of the edited output images into three key dimensions: **instruction reasoning**, **appearance consistency**, and **generation plausibility**. Evaluations are conducted using both human judges and an LMM-as-a-judge framework. For the latter, a rigorous pipeline was developed to ensure the reliability and validity of the LMM's assessments. Additionally, we performed extensive experiments to quantify the correlation between the scores produced by the LMM and human experts, which verifies the reliability and effectiveness of our proposed framework.

Using RISEBench, we conduct a systematic evaluation of state-of-the-art LMMs with visual editing capabilities. Our results reveal that open-source visual editing models such as BAGEL [8], Step1X-Edit [23], FLUX [19], EMU2 [35], and OmniGen [46] show limited reasoning capabilities, resulting in notably low performance across most test cases. Proprietary models, such as Gemini-2.0-Flash Series [38] and GPT-image-1, achieve significantly better overall performance. Notably, GPT-image-1 displays strong capabilities across temporal, causal, and spatial reasoning tasks. However, it still struggles with logical reasoning, highlighting an area for future research.

In summary, our main contributions are as follows:

1. We propose the first dedicated benchmark for assessing **R**easoning-**I**nformed vi**S**ual **E**diting (RISE), establishing a foundation for systematic assessment in this emerging area.
2. We define core categories of RISE challenges, design meaningful evaluation dimensions, and present a robust, effective LMM-as-a-judge framework for scalable and automated assessment.
3. We conduct a comprehensive evaluation and analysis of 8 prominent visual editing models, offering novel insights into their reasoning-driven visual editing capabilities and highlighting areas for future improvement.

---

*The Gemini-2.0-Flash Series comprises two models: Gemini-2.0-Flash-Experimental-Image-Generation (Gemini-2.0-Flash-Exp) and Gemini-2.0-Flash-Preview-Image-Generation (Gemini-2.0-Flash-Pre).

## 2  Related Work

**Image Editing with Diffusion Models.** Editing images based on textual user instructions is a crucial task in the field of image generation. With the advancement of large-scale diffusion models, the performance of image editing tasks has significantly improved. For instance, some methods [7, 25, 50, 33], adopt training-free approaches to guide denoising according to editing instructions, such as reversing noise on an image and guiding denoising with text [27], controlling attention maps during diffusion steps [11], or blending the original and generated images [6]. Recently, other works [4, 51, 49] have shifted to training-based methods, where pre-trained text-to-image diffusion models are further fine-tuned using datasets comprising paired edited images to enhance editing capabilities, yielding superior performance. However, due to the limited fine-grained semantic understanding of diffusion models, image editing models based on diffusion are often insufficient for handling complex, fine-grained editing instructions that require higher-order reasoning, thereby restricting their application in more diverse scenarios.

**Unified Large Multi-Modality Models.** Large Multi-Modality Models (LMMs) extend the input and output capabilities of large language models (LLMs) by incorporating visual information. Early works primarily focused on visual understanding, which involves processing visual inputs, reasoning, and generating textual outputs. Recently, a series of studies [34, 53, 9, 37, 40, 42, 45, 52, 48, 21] have aimed to develop unified LMMs capable of simultaneously handling both textual and visual inputs, enabling cross-modal generation and understanding. Initial approaches [34, 53, 9] often relied on pre-trained diffusion decoders to generate outputs by regressing CLIP [30] image representations. To further integrate understanding and generation, recent models such as Chameleon [37], Emu3 [40], and SynerGen-VL [20] have adopted a unified next-token prediction paradigm by discretizing images. Transfusion [52] and Show-o [48] demonstrated that bidirectional image diffusion could be integrated with autoregressive text prediction within a single framework.

**Text-to-Image Generation Evaluation**. The comprehensive evaluation of text-to-image generation is a long-standing problem. Early work mainly adopt the Fréchet inception distance (FID) [13] metric to mesure the distance between the generated distribution and the target distribution. However, this cannot measure the per-image alignment of image and the instruction. To better measure the semantic alignment in text-to-image generation, a series of works [10, 15, 12, 44, 43] propose metrics based on foundation models such as CLIP or object detectors. However, few works have focused on the reasoning-based visual editing. Recent work [28] shares the similar motivation of measuring models' world knowledge. However, they do not explicitly measure the models' reasoning

## 3  RISEBench-360

Humans possess a deep, conceptual understanding of objects and scenes in the real world that goes far beyond superficial attributes such as color and shape. For example, people can effortlessly reason about: 1) Temporal evolution of objects (**temporal reasoning**), such as fruits rotting over time, iron tools rusting, or children growing into adults; 2) Transformative changes due to external factors (**causal reasoning**), like ice cream melting under sunlight or vehicles becoming damaged after collisions; 3) Spatial configurations (**spatial reasoning**), including how shapes appear from different viewing angles and how various components assemble into complete structures. 4) Additionally, people can easily solve visual puzzle problems (**logical reasoning**) such as tic-tac-toe or mazes, and concretely imagine their solutions.

However, these capabilities present significant challenges for most generative models, which struggle to incorporate such reasoning into their visual outputs. To objectively assess current models' performance on these tasks and clearly identify their limitations, we propose RISEBench, the first benchmark specifically designed to evaluate reasoning-informed visual editing capabilities of image generative models across these dimensions of human-like visual understanding.

### 3.1  Benchmark Construction

Among the broad spectrum of visual editing tasks, RISEBench targets four major problem categories that require both deep visual understanding and precise reasoning, termed as *Temporal*, *Causal*, *Spatial*, and *Logical Reasoning*. For each category, we curate a diverse set of high-quality, carefully

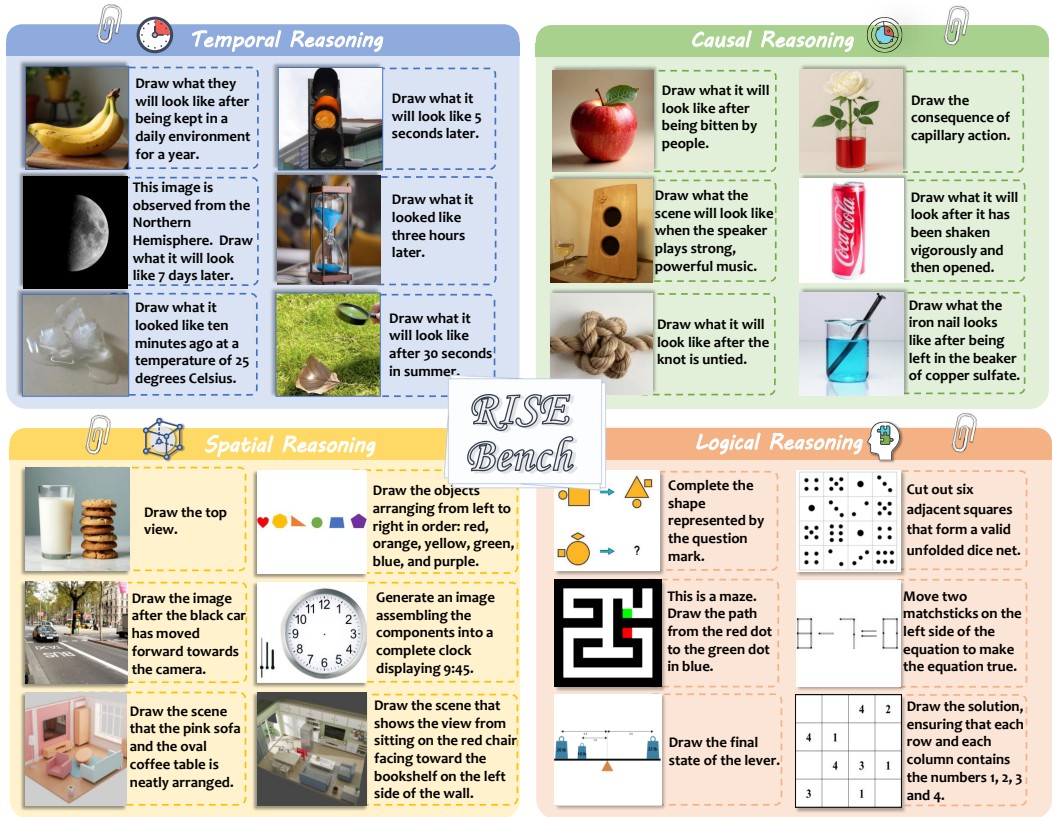

Figure 2: **Overview of RISEBench.** We present illustrative example questions from each of the four problem categories, each demanding profound image understanding and reasoning capabilities.

designed test cases. Each instance comprises an input image and an instruction prompt, illustrating reasoning-driven image transformations (see Fig. 2). The distribution of tasks is presented in Fig. 3.

**Temporal Reasoning.** Temporal reasoning tasks evaluate a model's ability to understand and antici­pate the evolution of objects or scenes over time. Beyond recognizing static attributes such as color, shape, or size, a reasoning-capable generative model should capture how these properties change through natural temporal progression. To construct such tasks systematically, we define several key elements of temporal change, including scale, direction, and object. Based on representative combinations of these dimensions, we derive four subcategories reflecting common temporal phenom­ena: *life progression*, *environmental cycles*, *material state change*, and *societal transformation*. The subcategories span diverse scenarios of temporal change, enabling the design of tasks that demand fine-grained understanding of temporal dynamics and assess a model's ability to perform temporally grounded reasoning beyond superficial image manipulation.

**Causal Reasoning.** Causal reasoning is essential for evaluating a generative model's ability to capture real-world interaction dynamics. Unlike temporal reasoning, which concerns natural progression over time, causal reasoning involves understanding how external forces or events directly induce changes in an object's state. This domain includes a range of phenomena: *1) Structural Deformation*, where external forces alter an object's shape; *2) State Transition*, such as phase changes (*e.g.*, freeze) triggered by manipulation or environmental shifts; *3) Chemical & Biological Transformations*, involving changes at the molecular or biological level; *4) Physical Manifestations*, where observable effects result from underlying physical laws activated by specific stimuli. These tasks require models to exhibit implicit knowledge of material properties, physical principles, and typical cause-effect relationships.

**Spatial Reasoning.** Spatial reasoning tasks assess a model's ability to understand, manipulate, and generate images that preserve accurate spatial relationships among objects in a scene. This requires internalizing geometric principles, structural coherence, 3D reasoning, and perspective —- core components of human-like visual understanding. We define five representative subcategories: *1)*

*Component Assembly* tests whether disjoint parts can be combined into a coherent whole, requiring spatial and structural integration; *2) Object Arrangement* evaluates the sequencing and positioning of objects based on attributes such as size, shape, or color; *3) Viewpoint Generation* assesses the ability to synthesize novel views from different angles, relying on latent 3D representations;

*4) Structural reasoning* challenges the model to complete occluded or fragmented objects by inferring missing parts; *5) Layout reasoning* examines understanding and manipulation of spatial configurations within a scene. Together, these tasks provide a comprehensive testbed for evaluating a model's spatial intelligence and its capacity for structure-aware, visually grounded generation.

**Logical Reasoning.** Unlike other categories that focus on physical or commonsense understanding in natural images, logical reasoning tasks evaluate a model's ability to perform structured, rule-based inference grounded in visual input. These tasks require interpreting visual elements and systematically applying formal rules — an area where current generative models still struggle. To assess this capability, we curate a diverse set of puzzles and logical challenges across three primary subtasks: *1) Puzzle Solving*, including classic visual problems such as Sudoku, mazes, and Tic-Tac-Toe; *2) Mathematical Derivation*, involving tasks requiring computation, such as shortest path finding and formula-based reasoning;

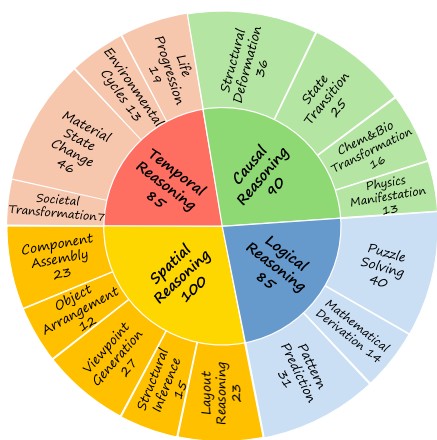

Figure 3: **Task Distribution of RISEBench.** RISEBench contains four main reasoning categories: *Temporal*, *Causal*, *Spatial*, and *Logical*. Each category includes various subtasks, facilitating a comprehensive evaluation.

*3) Pattern Prediction*, where the model must infer and complete visual patterns based on implicit rules. This category offers a broad spectrum of logic-based tasks with varying abstraction and difficulty, providing a rigorous evaluation of a model's visual-symbolic reasoning and its ability to link perception with inference.

## 3.2 Evaluation Pipeline

Evaluating the quality of reasoning-informed visual editing remains a challenging task. To address this, we first establish detailed scoring guidelines and conduct comprehensive human evaluations along three key dimensions: **1. Instruction Reasoning**, assessing whether the model correctly interprets and follows the editing instruction; **2. Appearance Consistency**, evaluating preservation of relevant visual attributes from the original image; **3. Visual Plausibility**, determining whether the output is coherent, realistic, and physically or logically plausible within context. Since human evaluation is resource-intensive and difficult to scale. To overcome these limitations, we further adopt an LMM-as-a-Judge approach. Given their strong visual understanding and reasoning abilities, state-of-the-art LMMs offer a promising alternative for automatic and human-aligned evaluation. We develop a robust evaluation pipeline (Fig. 4) leveraging these models to produce scalable assessments. In the following part, we detail each evaluation dimension:

*Dimension 1: Instruction Reasoning.* This dimension assesses the model's ability to accurately understand and execute the given instruction, with particular attention to both explicit directives and implicit requirements embedded within the prompt. A high-quality response not only performs the literal task specified but also captures the underlying reasoning or intended visual effect implied by the instruction. To improve the accuracy of LMMs in assessing instruction reasoning scores, we propose two evaluation methods. First, for samples with simple scenes that are easily describable through comprehensive text, we annotate a *reference text*, which serves as the ground truth and is used by the LMM to determine if the output image aligns with this description. For samples involving more complex scenes or unique shapes that are difficult to describe in text, particularly in Logical Reasoning and Spatial Reasoning tasks, we provide a *reference image* that fully matches the desired output. The judging LMM then compares the output image with the reference image to assess whether the instruction has been correctly executed. This approach expands the range of instruction types and ensures that the LMM can provide an accurate judgment score, with further details illustrated

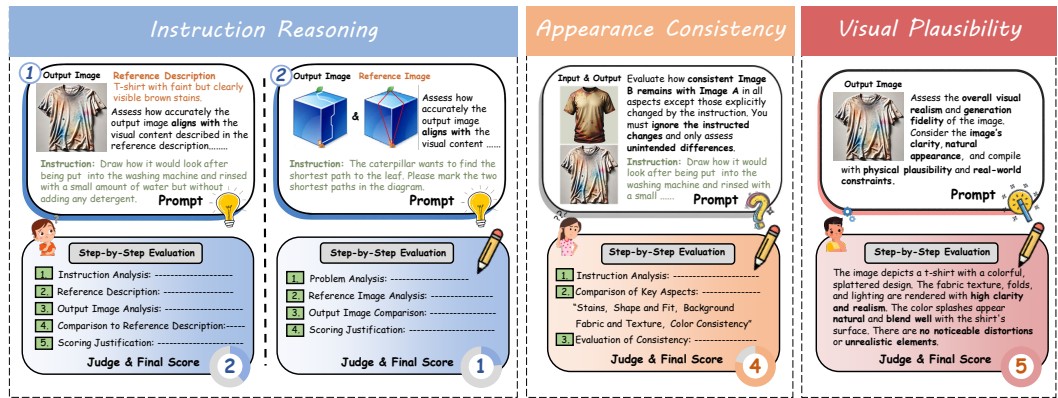

Figure 4: **Evaluation metrics of RISEBench.** RISEBench assesses the quality of generated images along three key dimensions: *Instruction Following*, *Appearance Consistency*, and *Visual Plausibility*. For each dimension, carefully crafted prompts are provided to the evaluator model (GPT-4.1 in this study), which analyzes various inputs and returns scores for each corresponding sub-dimension.

in Fig. 4(left). A full score (e.g., 5) is reserved for outputs that satisfy both the literal placement and the expected magnification, indicating robust instruction comprehension and reasoning.

*Dimension 2: Appearance Consistency.* Appearance consistency measures how well the visual elements unrelated to the instruction are preserved between the input and output images. This is particularly important in visual editing tasks, as it distinguishes between models that perform grounded edits based on the original image (*e.g.*, native generation models) and those that regenerate scenes from scratch (*e.g.*, cascade-based models). The LMM evaluates this metric by comparing the output image with the input image in accordance with the given instruction, as illustrated in Fig. 4(middle). For tasks involving temporal, causal, or spatial reasoning – where the input is typically a natural image rich in visual complexity – appearance consistency is scored on a continuous scale from 1 to 5, allowing for nuanced evaluation of how well the core scene is preserved post-editing. In contrast, logical reasoning tasks often involve stylized or synthetic inputs with simple layouts. Given their minimalistic structure, consistency in these cases is evaluated using a binary scheme: a score of 5 indicates full preservation of visual properties, while 1 reflects major deviations. This dimension ensures that models not only generate correct content but also do so in a way that respects the visual fidelity of the original input, which is essential for coherent and context-preserving visual editing.

*Dimension 3: Visual Plausibility.* The visual quality and realism of the generated image are critical factors in evaluating the performance of generative models. This dimension assesses whether the output is free from common generation artifacts such as blurriness, unnatural distortions, structural incoherence, or violations of physical laws. A plausible image should not only align with the instruction but also maintain visual integrity and realism consistent with how similar scenes would appear in the real world. We prompt the LMM to assess whether there are any implausible elements in the output image, as depicted in Fig. 4(right). The dimension only applies to tasks involving temporal, causal, or spatial reasoning – where outputs are expected to resemble natural images – visual plausibility is evaluated on a graded scale from 1 to 5, allowing for nuanced differentiation between high-quality and flawed generations. This dimension ensures that, beyond correctness and consistency, the generated images meet a basic threshold of visual fidelity and realism, which is essential for practical deployment of generative models in real-world applications.

The evaluation details, such as the specific instructions provided to judges (human evaluators and LMM-based assessors), carefully selected in-context examples, and the detailed configuration of the LMM judgement, are provided in Appx. H.

During evaluation, all dimension scores are normalized to the range [1, 5]. A sample is considered successfully solved only if it achieves scores of 5 on the three metrics, indicating full satisfaction of all applicable evaluation dimensions. *Accuracy* is then defined as the percentage of samples that are successfully solved out of the total number of test cases. The two complementary metrics offer both fine-grained performance measurement and an interpretable success rate across tasks.

Table 1: **Overall performance on RISEBench-360.** GPT-image-1 achieves the highest performance with an accuracy of only 28.9%, followed by Gemini-2.0-Flash Series with the second-highest and third-highest accuracy. The remaining models perform close to zero, highlighting the significant challenges that remain in achieving robust reasoning-informed visual editing.

| Models | Temporal | Causal | Spatial | Logical | Overall |
|---|---|---|---|---|---|
| GPT-image-1 [16] | **34.1%** | **32.2%** | **37.0%** | **10.6%** | **28.9%** |
| Gemini-2.0-Flash-exp [38] | 8.2% | 15.5% | 23.0% | 4.7% | 13.3% |
| Gemini-2.0-Flash-pre [38] | 10.6% | 13.3% | 11% | 2.3% | 9.4% |
| BAGEL [8] | 3.5% | 4.4% | 9.0% | 5.9% | 5.8% |
| Step1X-Edit [24] | 0.0% | 2.2% | 2% | 3.5% | 1.9% |
| OmniGen [46] | 1.2% | 1.0% | 0.0% | 1.2% | 0.8% |
| EMU2 [35] | 1.2% | 1.1% | 0.0% | 0.0% | 0.5% |
| HiDream-Edit [14] | 0.0% | 0.0% | 0.0% | 0.0% | 0.0% |
| FLUX.1-Canny [19] | 0.0% | 0.0% | 0.0% | 0.0% | 0.0% |

## 4 Experiments

To evaluate the performance of representative visual editing approaches, we selected a diverse set of models encompassing various architectures and generation paradigms. Specifically, **Flux1.0-Canny** [19] serves as a representative diffusion-based editing model; **EMU2** [35], **OmniGen** [46] and **BAGEL** [8] exemplify the auto-regressive generation paradigm; and **Step1X-Edit** [24] represents a hybrid model that combines a LMM with a DiT-style diffusion architecture. We also include four proprietary models: **HiDream-Edit** [14]. **Gemini 2.0-Flash-Preview** [38], **Gemini 2.0-Flash-Experimental** [38], and **GPT-image-1** [16]. For all of the proprietary models, we obtained their outputs directly via their respective official API service.

### 4.1 Main Results (LMM-as-a-Judge)

We report the accuracy performance on a 100-point scale in Tab. 1, with representative output examples shown in Fig. 6. All scores are assigned by the GPT-4.1 model, which serves as the judger in our LMM-as-a-Judge evaluation pipeline.

Among the evaluated models, the recently released GPT-image-1 demonstrates the highest performance on RISEBench. **However, its accuracy of 28.9% remains relatively low, highlighting persistent limitations in performing the complex visual reasoning required for these editing tasks.** Following GPT-image-1, Gemini-2.0-Flash-Experimental and Gemini-2.0-Flash-Preview rank second and third, respectively. Gemini-2.0-Flash-Experimental achieves an average score of 13.3%, while Gemini-2.0-Flash-Preview reaches an accuracy of 9.4%. Notably, although Gemini-2.0-Flash-Preview exhibits superior image generation quality compared to Gemini-2.0-Flash-Experimental, it appears to suffer a significant decline in spatial reasoning capabilities (accuracy dropping from 23.0% to 11.0%), resulting in a lower overall performance. In stark contrast, other models, including Step1X, OmniGen, EMU2, FLUX.1-Canny, and HiDream, all exhibit significantly poor performance on the RISEBench. Their accuracy scores are all close to 0%, indicating limited understanding of the input images and a failure to generate semantically meaningful edits.

In temporal, causal, and spatial reasoning tasks, where input images typically depict natural scenes and instructions often emphasize common knowledge, GPT-image-1 demonstrates strong performance, with accuracies exceeding 30%. However, when confronted with logical reasoning tasks involving complex logical puzzles and intricate instructions, **GPT-image-1 encounters significant challenges, achieving only an accuracy of 10.6%**. This disparity underscores logical reasoning as a critical bottleneck, representing a crucial avenue for future research in reasoning-guided visual generation.

To gain deeper insights into the strengths and limitations of each model, we analyze the average performance across three evaluation dimensions for the evaluated models, as illustrated in Fig. 5. The results indicate that GPT-image-1 achieves significantly leading performance across all three evaluation metrics: Instruction Reasoning, Appearance Consistency, and Visual Plausibility. This positions it as the most powerful model among those evaluated for reasoning-based editing tasks.

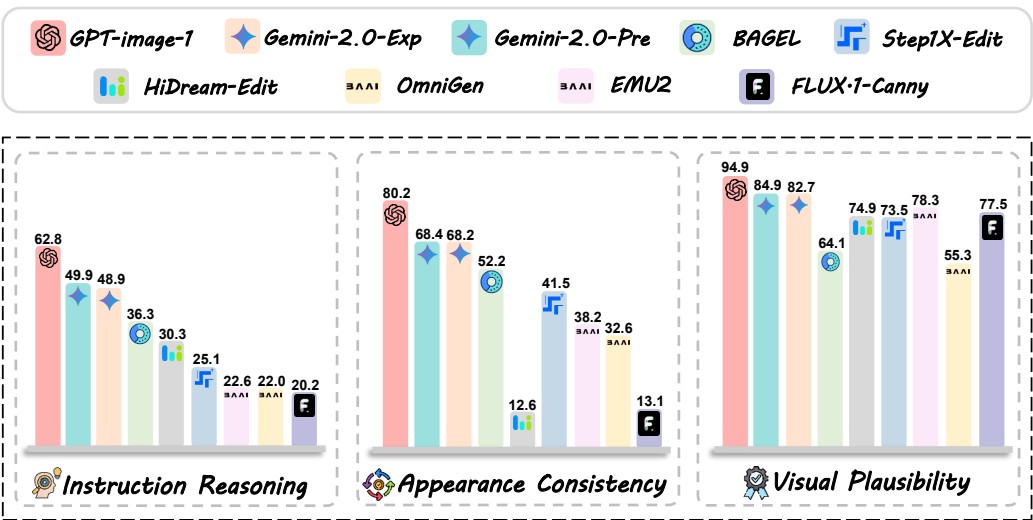

Figure 5: **Comparison across models on three evaluation sub-dimensions.** GPT-image-1 demonstrates superior performance, achieving the highest scores across all three evaluation metrics. Gemini-2-Flash-Series also exhibits competitive performance on these criteria. In contrast, the performance of many other evaluated models was considerably lower, indicating significant limitations in their ability to follow instructions and maintain visual integrity.

The Gemini-2.0-Flash models (experimental and preview versions) exhibit a minor difference in performance; both demonstrate relatively high scores across the three metrics, resulting in the second-best overall performance. This suggests they possess some capability in understanding complex instructions.

BAGEL also demonstrates a degree of understanding capability, as reflected by its performance in Instruction Reasoning and Appearance Consistency, albeit with scores lower than those of the Gemini Series. However, its Visual Plausibility score is notably low, ranking as the second lowest among the evaluated models. This indicates a potential strength in semantic understanding coupled with a weakness in the image generation process. In contrast, the other three models all lack sufficient capability in the reasoning-informed visual editing task. Among these five models, HiDream-Edit demonstrates the best performance in Instruction Reasoning; however, it also exhibits the lowest score in Appearance Consistency, indicating an inability to maintain the characteristics of the main content. Step1X achieves a score of 41.5 in Appearance Consistency but lacks the ability to understand instructions, positioning it as a standard editing model. Both EMU2 and OmniGen demonstrate similarly limited performance in Instruction Reasoning and Appearance Consistency. However, OmniGen's performance in Visual Plausibility is markedly poorer, exhibiting the lowest score among the evaluated models. This suggests a notable weakness in OmniGen's underlying image generation capability Regarding FLUX.1-Canny, it shows poor performance in both understanding instructions and maintaining appearance consistency, demonstrating significantly limited performance. The complete score distributions are presented in Appx. D.

## 4.2 Analysis for Models

We exhibit several representative model outputs in Fig. 6 and observe several notable characteristics of the evaluated models. First, GPT-image-1 demonstrates substantial robustness in visual editing tasks. Beyond its proficient instruction comprehension, a critical attribute is its ability to preserve the original image content even when faced with ambiguous or misunderstood instructions (Fig. 6, Temporal [2], Spatial [1], Logical [1,2]). This behavior directly contributes to its superior performance in terms of Appearance Consistency and Visual Plausibility.

In contrast, the Gemini-2.0-Flash Series exhibits a comparatively limited capacity for instruction understanding relative to GPT-image-1. It frequently introduces artifacts by either adding extraneous elements or omitting critical content during the editing process (Fig. 6, Temporal[1], Spatial[2]), thereby diminishing image consistency. Moreover, when instructions are entirely misinterpreted,

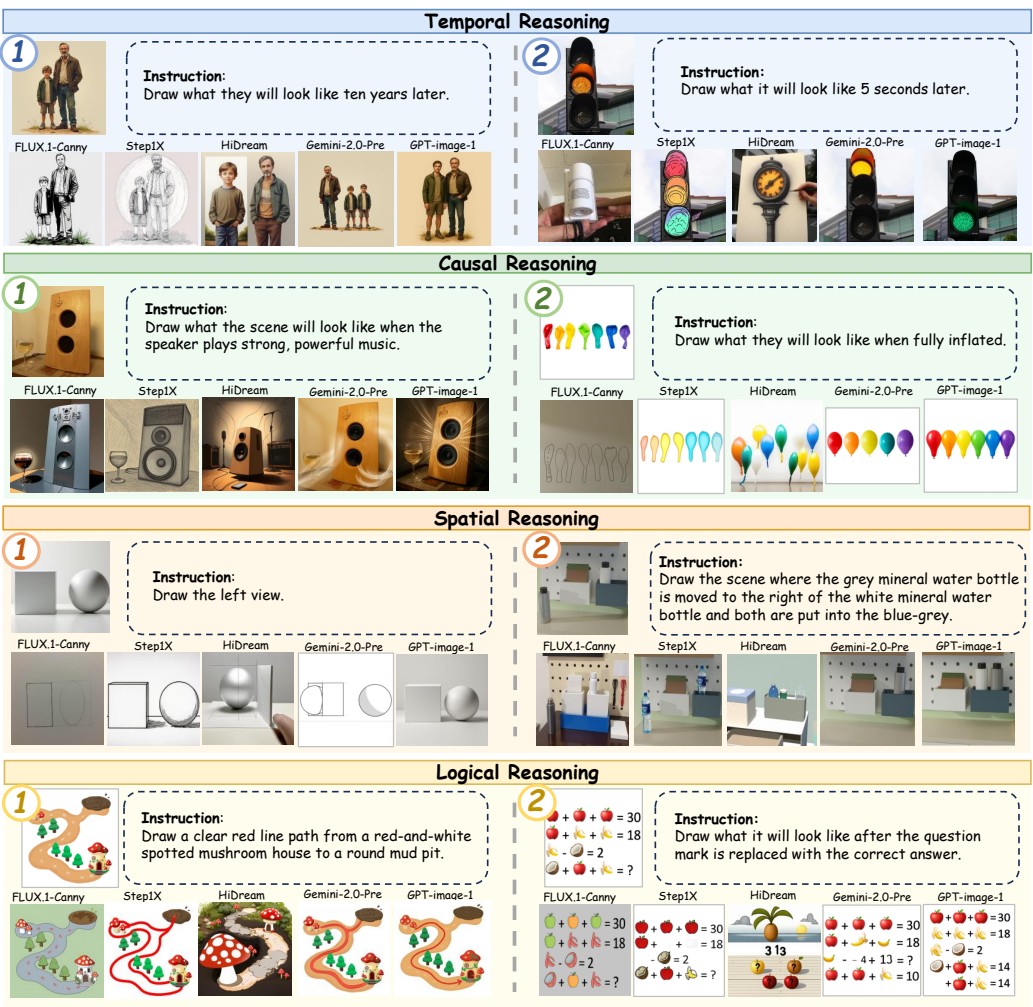

Figure 6: **Examples of several different models' outputs on RISEBench-360.** The analyzed models demonstrate distinct characteristics in their responses. Specifically, GPT-image-1 exhibits instances of instruction misunderstanding, while Gemini sometimes struggles with maintaining image consistency. Other models generally show limited ability to comprehend and execute complex instructions.

Gemini-2.0-Flash-preview tends to generate chaotic or severely distorted reconstructions (Fig. 6, Spatial[1], Logical[2]), leading to significantly degraded output quality.

Regarding the remaining models, HiDream-Edit displays a weak understanding of certain instructions but often yields unconventional or anomalous image reconstructions. Step1X-Edit and Flux.1-Canny appear largely restricted to processing instructions featuring explicit, concrete nouns, exhibiting minimal to negligible broader reasoning capabilities.

### 4.3 Validity of LMM-as-a-Judge

To assess the validity of using LMMs as evaluators, we analyze the correlation between LMM-based assessments and human expert judgments. We conduct the user study involving six human experts, who independently score the randomly sampled 100 outputs of two models (Gemini-2.0-Flash-Experimental and GPT-image-1) based on criteria aligned with those used in LMM-based evaluations. We analyzed the human expert scores corresponding to each score assigned by LMM-as-a-judge (on a scale of 1–5). For each assigned model score, we report the proportion of samples, mean, standard deviation (Std.), mean error, and Mean Absolute Error (MAE) of the corresponding human scores. Furthermore, we computed the overall MAE between the complete sets of scores provided by human experts and those assigned by LMM. These results are presented in Tab. 2.

Table 2: **Correlation between human and model-based judgments.** For each score level assigned by the model(1-5), we report the distribution of the corresponding human expert scores, along with their proportion, mean, standard deviation (Std.), and Mean Absolute Error (MAE). The overall MAE of the complete sets is also presented. *Reas.*, *Cons.* and *Plau.* denote Instruction Reasoning, Appearance Consistency and Visual Plausibility respectively.

| Model | Proportion | | | Human Mean | | | Human Std. | | | Mean Error | | | MAE | | |
|---|---|---|---|---|---|---|---|---|---|---|---|---|---|---|---|
| Score | Reas. | Cons. | Plau. | Reas. | Cons. | Plau. | Reas. | Cons. | Plau. | Reas. | Cons. | Plau. | Reas. | Cons. | Plau. |
| 1 | 27% | 1% | 0% | 1.1 | 2.6 | - | 0.1 | 0.0 | - | 0.1 | 1.6 | - | 0.1 | 1.6 | - |
| 2 | 11% | 5% | 0% | 2.2 | 3.3 | - | 1.0 | 0.6 | - | 0.2 | 1.3 | - | 0.1 | 1.3 | - |
| 3 | 10% | 13% | 12% | 3.6 | 3.6 | 4.1 | 1.2 | 0.5 | 0.7 | 0.6 | 0.6 | 1.1 | 1.2 | 0.7 | 1.2 |
| 4 | 13% | 9% | 9% | 4.6 | 4.3 | 4.6 | 0.5 | 0.4 | 0.2 | 0.6 | 0.3 | 0.6 | 0.8 | 0.4 | 0.6 |
| 5 | 39% | 61% | 79% | 4.7 | 4.7 | 4.8 | 0.4 | 0.4 | 0.3 | -0.3 | -0.3 | -0.2 | 0.3 | 0.3 | 0.2 |
| Overall | - | - | - | - | - | - | - | - | - | - | - | - | **0.5** | **0.7** | **0.4** |

The distribution indicates that the scores assigned by human experts are closely aligned with those predicted by the model, demonstrating a strong overall consistency. The MAE is consistently low across the evaluation dimensions. For the three primary evaluation criteria—Instruction Reasoning, Appearance Consistency, and Visual Plausibility—the observed MAEs were 0.5, 0.7, and 0.4 respectively, which are notably low relative to the 1-5 scoring scale, with each MAE falling below 1.

Leveraging the robust design of our evaluation pipeline, our LMM-as-a-Judge pipeline demonstrates effectiveness in identifying both high-quality outputs and significant failures. Specifically, the LMM-Judge assigns the max score (5) to a substantial proportion of outputs across all three evaluation metrics. For this subset of outputs rated 5 by LMM, the corresponding mean scores assigned by human experts were notably high (4.7, 4.7, and 4.8). Furthermore, MAE between LMM and the human scores for these outputs is low (only 0.3, 0.3, and 0.2). These findings collectively indicate strong agreement between LMM and human for outputs considered successful. Besides, LMM also exhibits proficiency in identifying outputs that critically fail to adhere to instructions. Specifically, the model assigned a Reasoning score of 1 to 27% of the samples. For this subset, the corresponding human expert mean score is 1.1, resulting in an MAE of merely 0.1. This demonstrates excellent agreement between the LMM and human in pinpointing outputs with significant reasoning deficiencies.

When the model assigns intermediate scores (specifically 2, 3), the alignment with human judgments tends to decrease. This reduced agreement is primarily attributable to the subjective nature of the scoring criteria, which inherently leads to greater variability and potential disagreements when evaluating the same sample, even among human experts. More specifically, for the Appearance Consistency and Visual Plausibility metrics, human experts demonstrated a tendency to assign higher scores compared to the model. This discrepancy may stem from the model's potentially more meticulous examination of the generated images, allowing it to identify subtle inconsistencies or deviations from the original content that human evaluators might overlook.

## 5 Conclusion

In this paper, we introduced RISEBench – the first dedicated benchmark for evaluating the Reasoning-Informed Visual Editing (RISE) capabilities of multimodal models. RISEBench targets four core types of reasoning: temporal, causal, spatial, and logical, and provides a structured evaluation framework that takes into account instruction reasoning, appearance consistency, and generation plausibility. Through extensive experiments, we observed that GPT-image-1 significantly outperform its open-source and proprietary counterparts. However, even the most advanced models continue to exhibit notable shortcomings in logical reasoning tasks, highlighting a key area for future research and model development.

## Acknowledgement

This work was partly supported by National Natural Science Foundation of China (62506229, 62171281), Natural Science Foundation of Shanghai (25ZR1402268), Science and Technology Commission of Shanghai Municipality (STCSM, Grant Nos. 20DZ1200203).

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

# A Comparison across models on three evaluation sub-dimensions

Table 3: **Comparison across models on three evaluation sub-dimensions.**

| Model | Instruction Reasoning | Appearance Consistency | Visual Plausibility |
|---|---|---|---|
| Gemini-2.5-Flash-Image[38] | 61.2 | **86.0** | 91.3 |
| GPT-Image-1[16] | **62.8** | 80.2 | **94.9** |
| GPT-Image-1-mini[16] | 54.1 | 71.5 | 93.7 |
| Gemini-2.0-Flash-exp[38] | 48.9 | 68.2 | 82.7 |
| BAGEL (w/ CoT)[8] | 45.9 | 73.8 | 80.1 |
| Seedream-4.0[32] | 58.9 | 67.4 | 91.2 |
| Gemini-2.0-Flash-pre[38] | 49.9 | 68.4 | 84.9 |
| Qwen-Image-Edit[41] | 37.2 | 66.4 | 86.9 |
| BAGEL[8] | 36.5 | 53.5 | 73.0 |
| FLUX.1-Kontext-Dev[2] | 26.0 | 71.6 | 85.2 |
| Ovis-U1[39] | 33.9 | 52.7 | 72.9 |
| HiDream-Edit[14] | 30.3 | 12.6 | 74.9 |
| Step1X-Edit[24] | 25.1 | 41.5 | 73.5 |
| EMU2[34] | 22.6 | 38.2 | 78.3 |
| OmniGen[46] | 22.0 | 32.6 | 55.3 |
| FLUX.1-Canny[19] | 20.2 | 13.1 | 77.5 |

Comparison of models across three evaluation sub-dimensions is shown in Table 3.

# B Data Source of RISEBench

Input images for the RISEBench dataset are primarily sourced from the following categories:

1. Images generated by image generation models.
2. Images rendered from 3D environments utilizing software(Blender).
3. Images derived from existing datasets and benchmarks [47, 36].
4. Images collected from the internet under permissive licenses.

# C Performance across Subtasks

Table 4: **Detail performance across subtasks within the four prominent categories.** GPT-4o-Image shows great capability in common scenarios, but it still struggles with complex tasks like Chemical, Biology and Physics tasks. Besides, while GPT-4o-Image exhibits relatively strong performance on the Mathematical Derivation subtask, its capability is notably diminished, approaching near-zero effectiveness, in subtasks like Pattern Prediction and Puzzle Solving.

| Subtask/Model | GPT-4o-Image | Gemini-Pre | Gemini-Exp | BAGEL | Step-1X | OmniGen | HiDream | EMU2 | FLUX.1 |
|---|---|---|---|---|---|---|---|---|---|
| *Temporal Reasoning* | | | | | | | | | |
| Life Progression | 52.6 | 0.0 | 5.3 | 0.0 | 0.0 | 0.0 | 0.0 | 0.0 | 0.0 |
| Material Progression | 32.6 | 15.2 | 6.5 | 4.3 | 0.0 | 0.0 | 0.0 | 0.0 | 0.0 |
| Environmental Cycles | 30.7 | 15.4 | 15.3 | 7.7 | 0.0 | 7.6 | 0.0 | 0.0 | 0.0 |
| Societal Transformation | 0.0 | 0.0 | 14.3 | 0.0 | 0.0 | 0.0 | 0.0 | 0.0 | 0.0 |
| *Causal Reasoning* | | | | | | | | | |
| Structural Deformation | 41.7 | 13.9 | 13.9 | 5.5 | 0.0 | 0.0 | 0.0 | 0.0 | 0.0 |
| State Transition | 36.0 | 20.0 | 20.0 | 4.0 | 0.0 | 0.0 | 0.0 | 0.0 | 0.0 |
| Chem&Bio Transform | 12.5 | 6.3 | 12.5 | 0.0 | 6.2 | 0.0 | 0.0 | 0.0 | 0.0 |
| Physics Manifestation | 23.0 | 7.7 | 15.4 | 0.0 | 0.0 | 0.0 | 0.0 | 0.0 | 0.0 |
| *Spatial Reasoning* | | | | | | | | | |
| Component Assembly | 56.5 | 26.1 | 26.1 | 13.0 | 0.0 | 0.0 | 0.0 | 0.0 | 0.0 |
| Object Arrangement | 25.0 | 8.3 | 8.3 | 0.0 | 0.0 | 0.0 | 0.0 | 0.0 | 0.0 |
| Viewpoint Generation | 44.4 | 11.1 | 44.4 | 11.1 | 3.7 | 0.0 | 0.0 | 0.0 | 0.0 |
| Structural Inference | 26.6 | 6.7 | 0.0 | 0.0 | 0.0 | 0.0 | 0.0 | 0.0 | 0.0 |
| Layout Reasoning | 21.7 | 0.0 | 17.4 | 13.0 | 4.3 | 0.0 | 0.0 | 0.0 | 0.0 |
| *Logical Reasoning* | | | | | | | | | |
| Pattern Prediction | 3.22 | 0.0 | 0.0 | 0.0 | 3.3 | 0.0 | 0.0 | 0.0 | 0.0 |
| Mathematical Derivation | 35.7 | 0.0 | 21.4 | 14.3 | 0.0 | 0.0 | 0.0 | 0.0 | 0.0 |
| Puzzle Solving | 7.5 | 5 | 2.5 | 7.5 | 0.0 | 2.5 | 0.0 | 0.0 | 0.0 |

The performance of the eight evaluated models across the subtasks within the four prominent categories is presented in Tab. 4. Analysis of this table reveals distinct patterns in model capabilities.

GPT-4o-Image, considered as the leading visual editing model, demonstrates strong proficiency in tasks requiring instruction understanding and execution within common scenarios, such as Life Progression, Structural Deformation, and Viewpoint Generation. However, its performance significantly declines when faced with less common or more complex scenarios, including Chemistry & Biology Transformation, Societal Transformation, and Physics Manifestation, as shown in Fig. 7. In these cases, the model struggles to produce consistently accurate edits. Furthermore, examining the Logical Reasoning category, which generally demands a higher level of complex understanding, reveals nuanced performance: While GPT-4o-Image exhibits relatively strong performance on the Mathematical Derivation subtask, its capability is notably diminished, approaching near-zero effectiveness, in subtasks like Pattern Prediction and Puzzle Solving. These findings, particularly the struggles in complex or domain-specific scenarios and certain logical reasoning tasks, further underscore the current limitations of state-of-the-art visual-editing models.

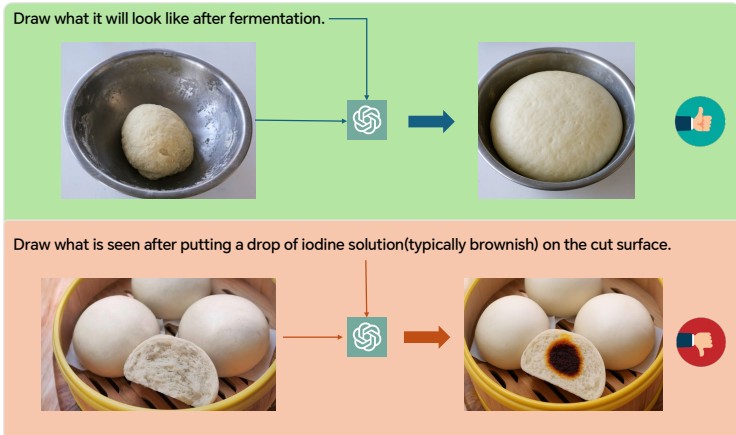

Figure 7: **GPT-4o-Image's Understanding Capabilities in different Tasks.** While GPT-4o-Image can effectively handle tasks in common scenarios, its performance declines significantly on tasks necessitating deeper or more difficult understanding.

## D   Score Distribution of Model Outputs

The score distribution of the eight evaluated models on the RISEBench benchmark is illustrated in Fig. 8. Analysis of these distributions reveals that GPT-4o-Image and the Gemini-Series models consistently achieve a high proportion of favorable scores across all three evaluation metrics: Instruction Reasoning, Appearance Consistency, and Visual Plausibility. In contrast, the performance of other models is notably weaker, particularly concerning instruction reasoning and appearance consistency, where they exhibit a low proportion of high scores. Furthermore, OmniGen specifically demonstrates significant difficulties in maintaining the visual plausibility of the generated images. This inability compromises the quality of its outputs and contributes to its comparatively lower overall performance on the benchmark.

## E   Interactive Interface for Human Annotators

A view of the user interface (UI) employed for human annotation is shown in Fig. 9.

## F   Limitations

As this is the first benchmark evaluating reasoning-informed image editing capabilities, our work is still in its initial stages. The categories of tasks included may not be exhaustive, and the dataset size, comprising only 360 questions, is not substantial.

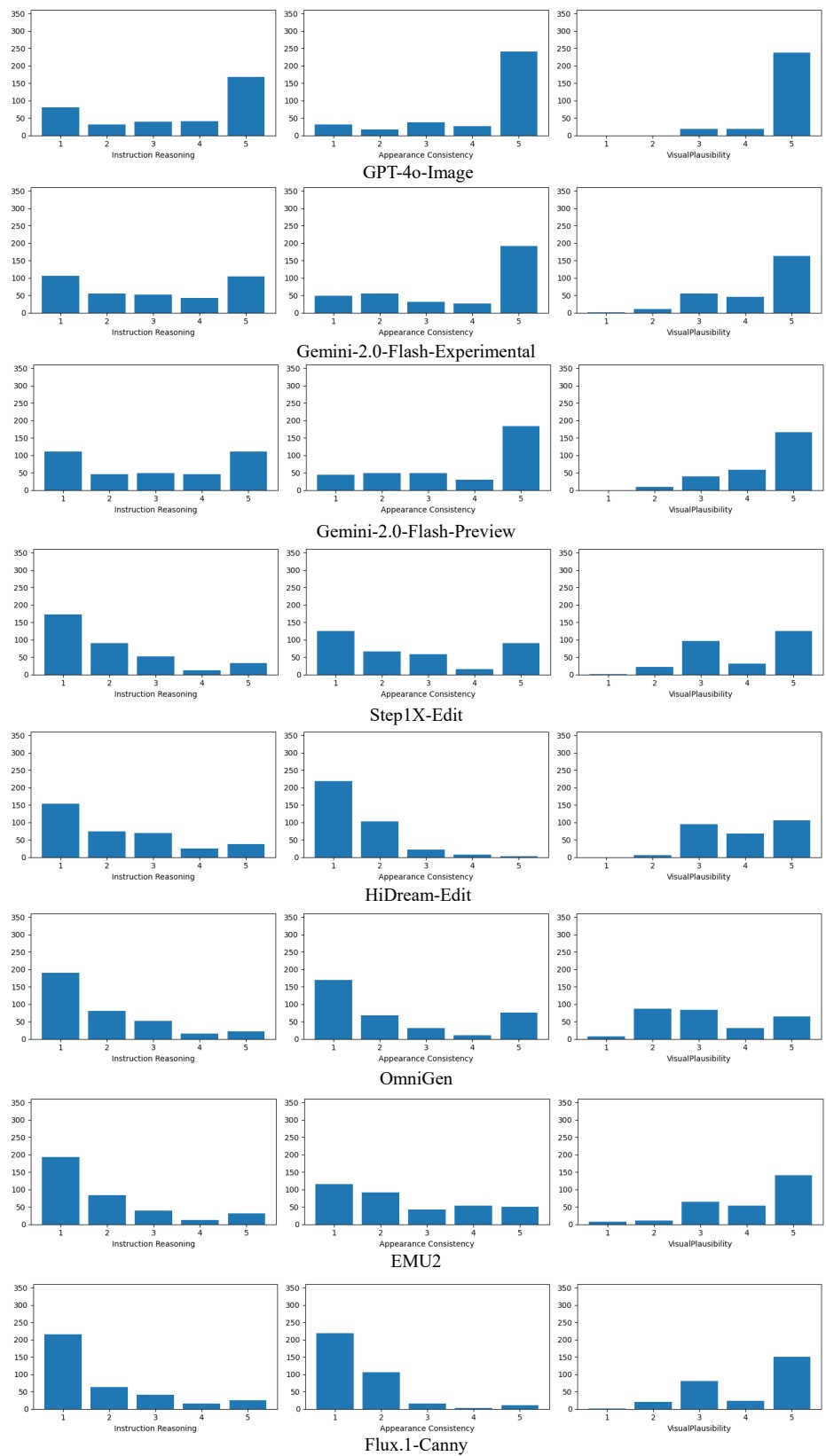

Figure 8: **The score distribution of the model being tested.**

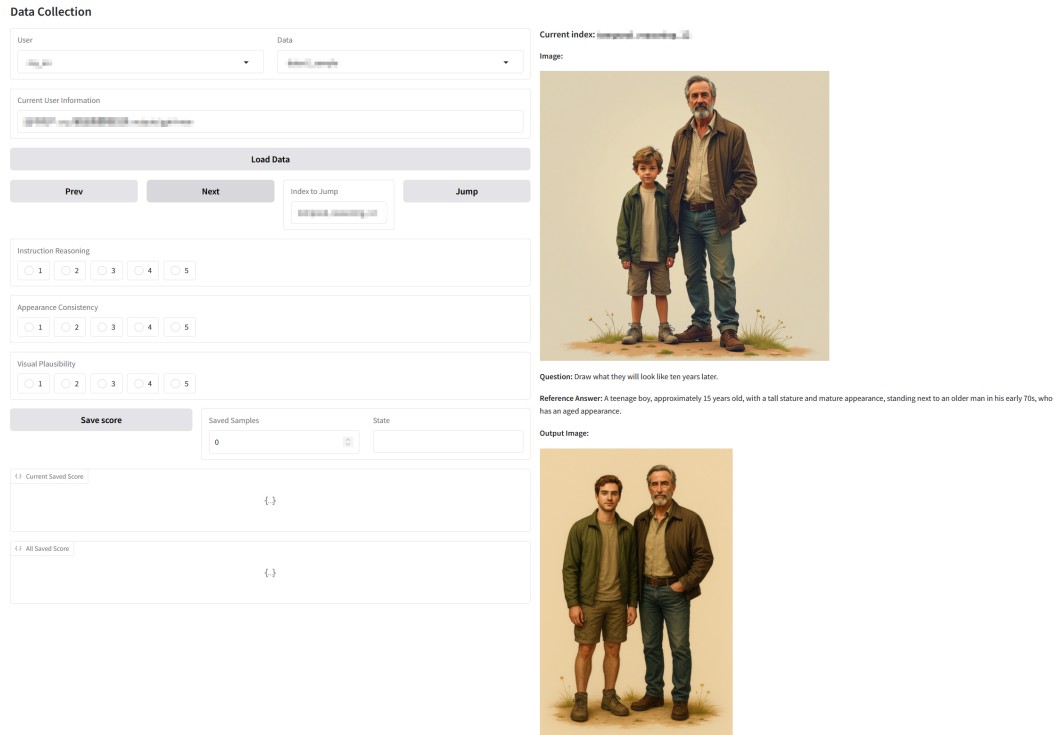

Figure 9: **Interactive Interface provided for Human Annotators.**

# G    Detailed Outputs of All Evaluated Models

The outputs of all evaluated models on our RISEBench benchmark are presented below for comprehensive comparison.

# H    Prompt for Judgement

We exhibit all our prompts for GPT-4o judger across different metrics and dimensions here.

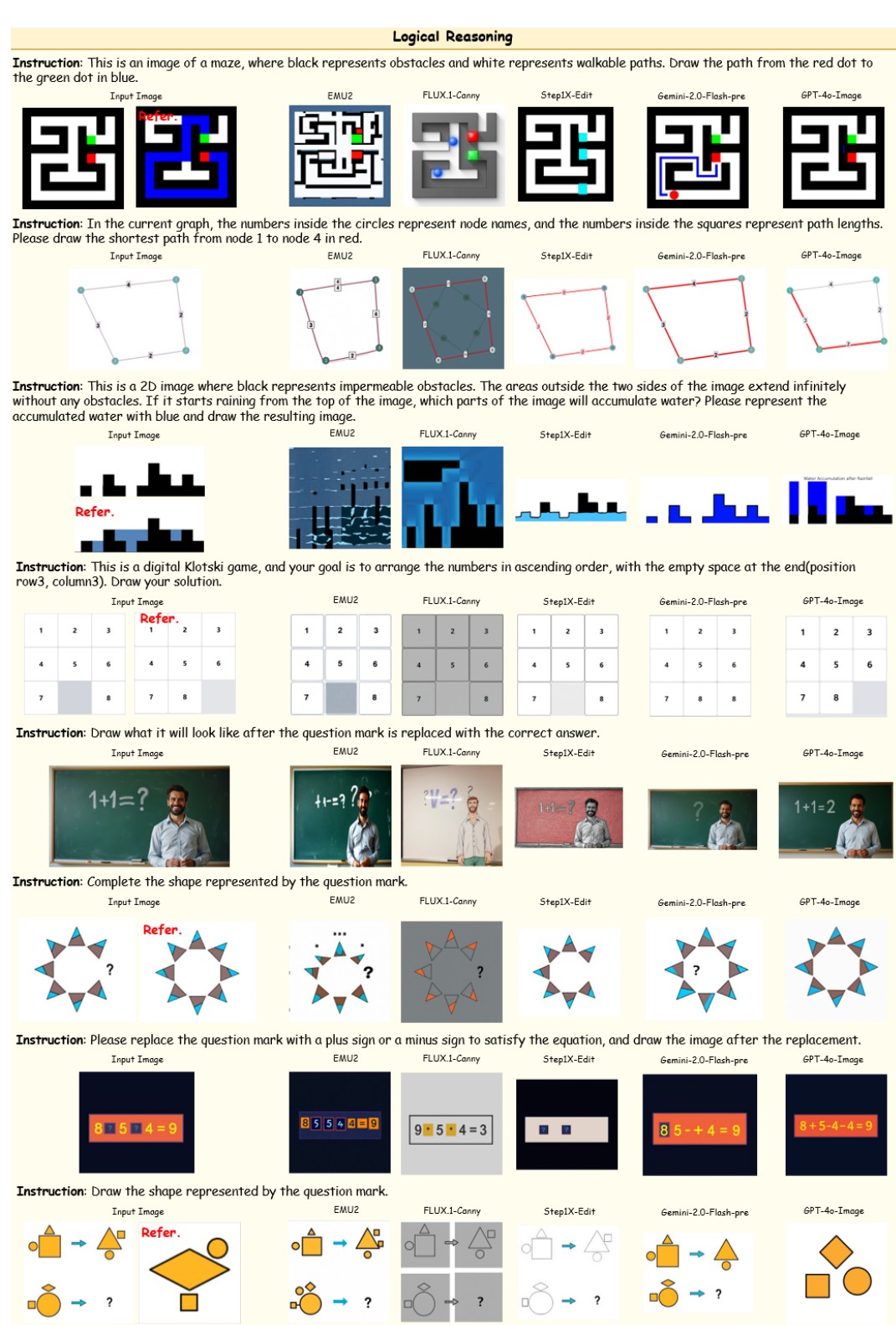

Figure 10: **Logical Reasoning Outputs – Part 1.**

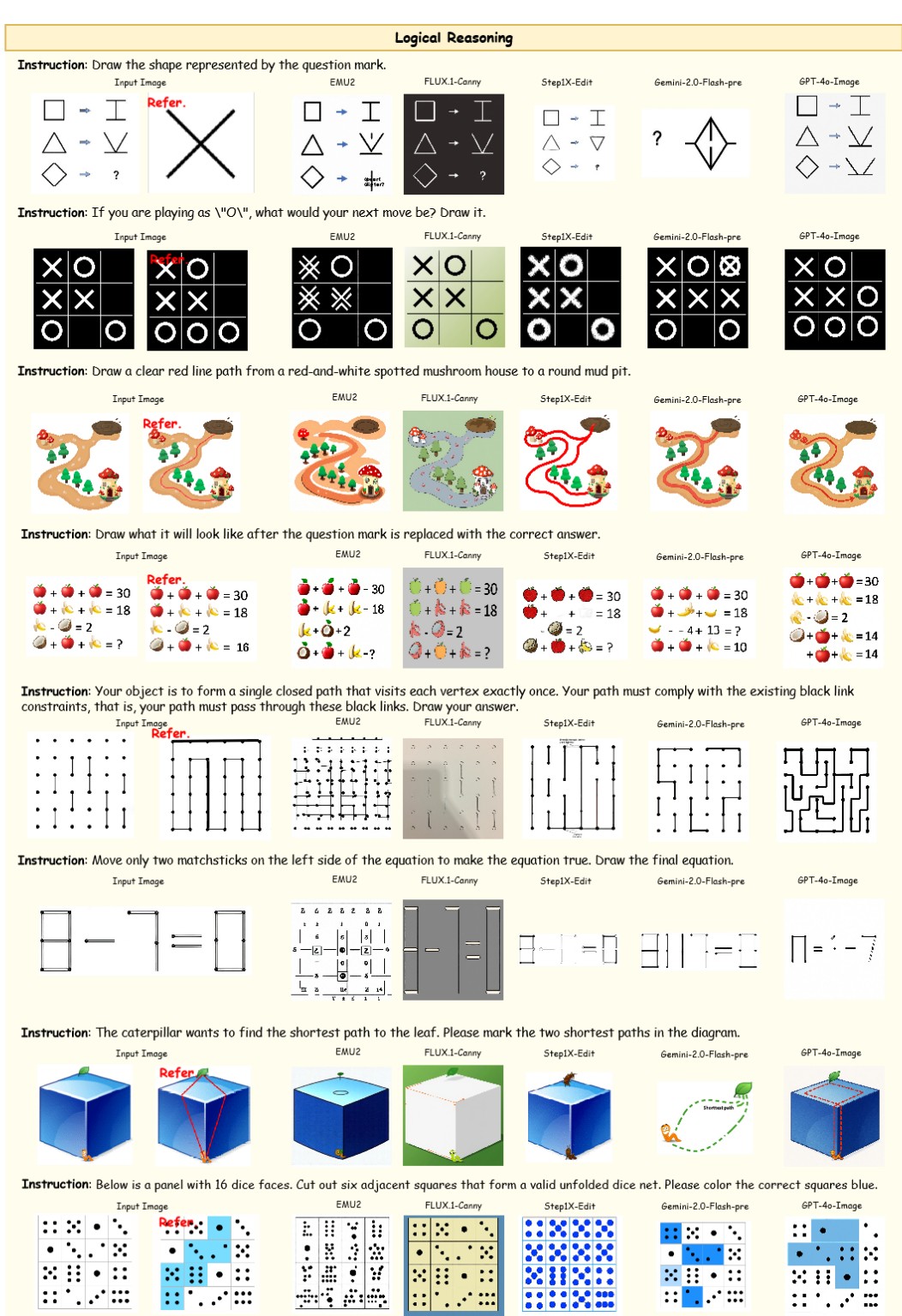

Figure 11: **Logical Reasoning Outputs – Part 2.**

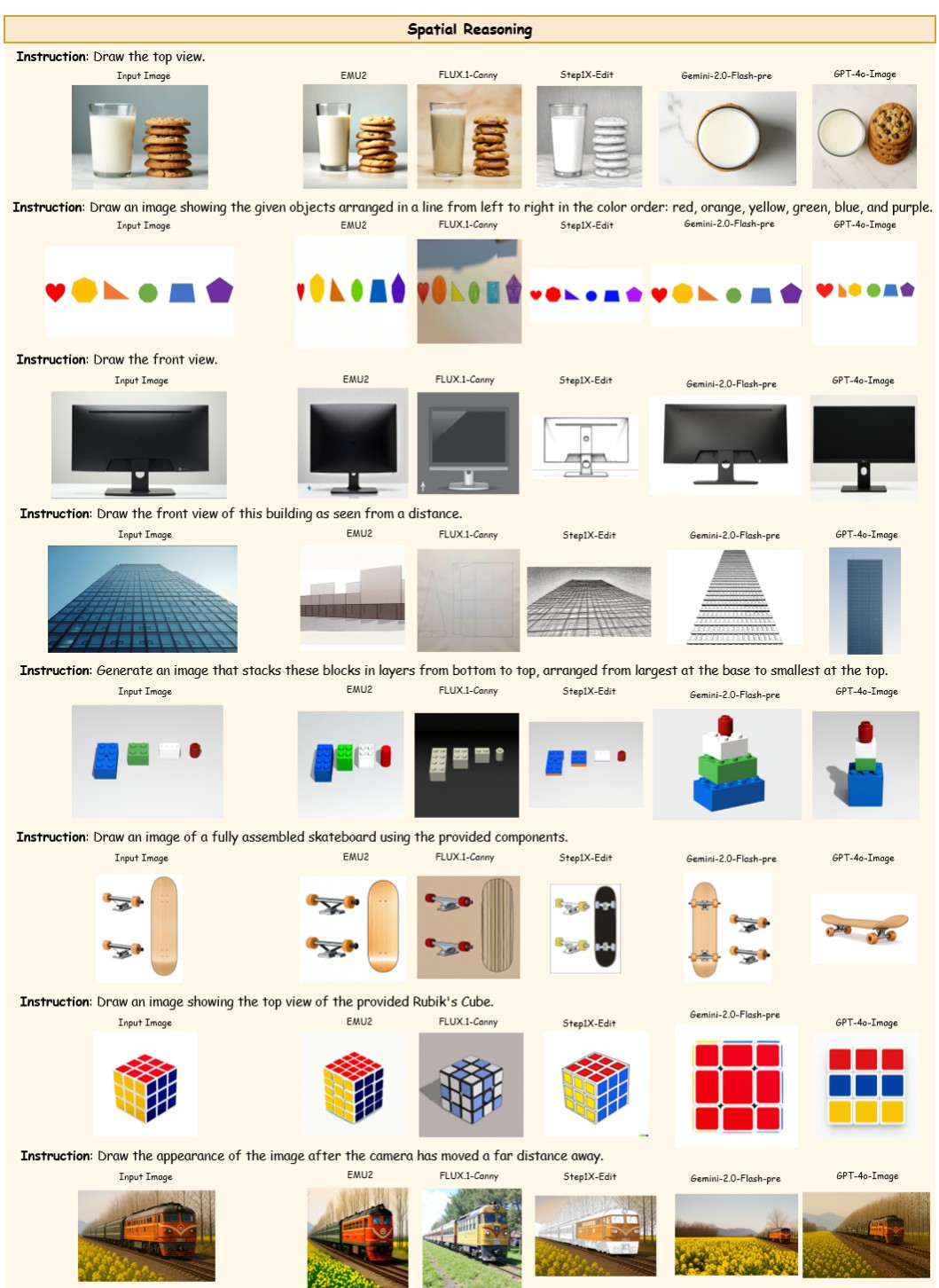

Figure 12: **Spatial Reasoning Outputs – Part 1.**

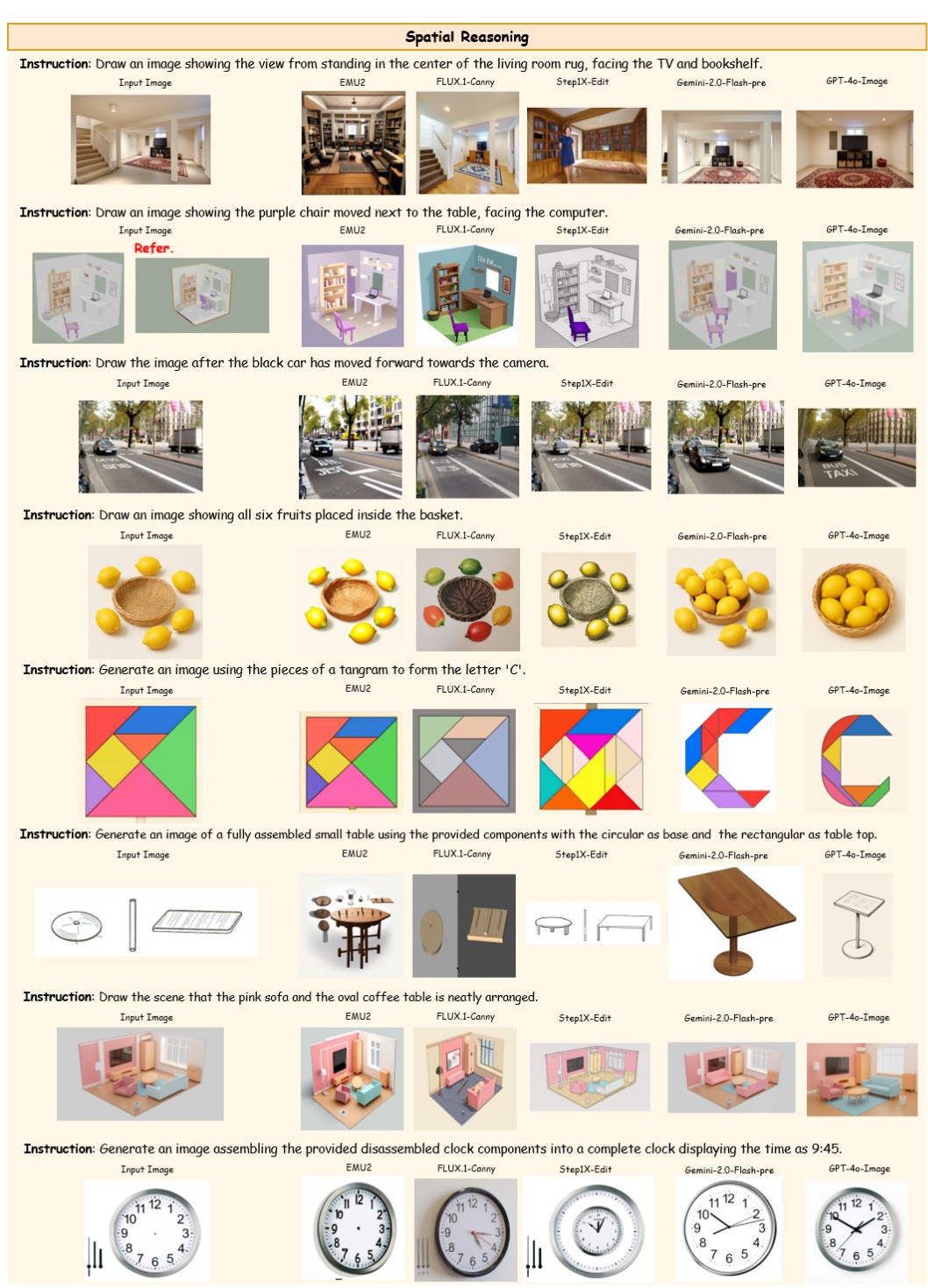

Figure 13: **Spatial Reasoning Outputs – Part 2.**

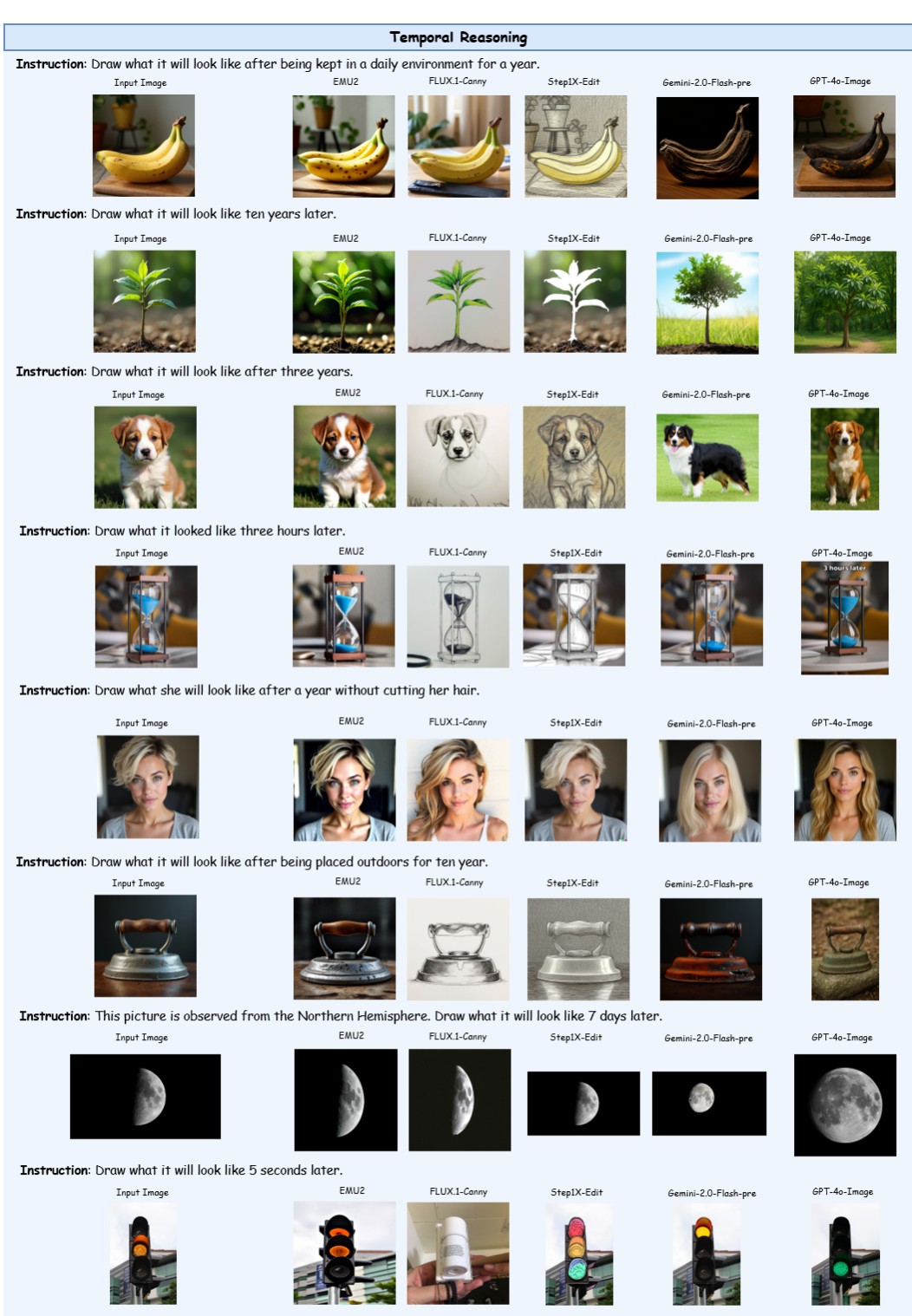

Figure 14: **Temporal Reasoning Outputs – Part 1.**

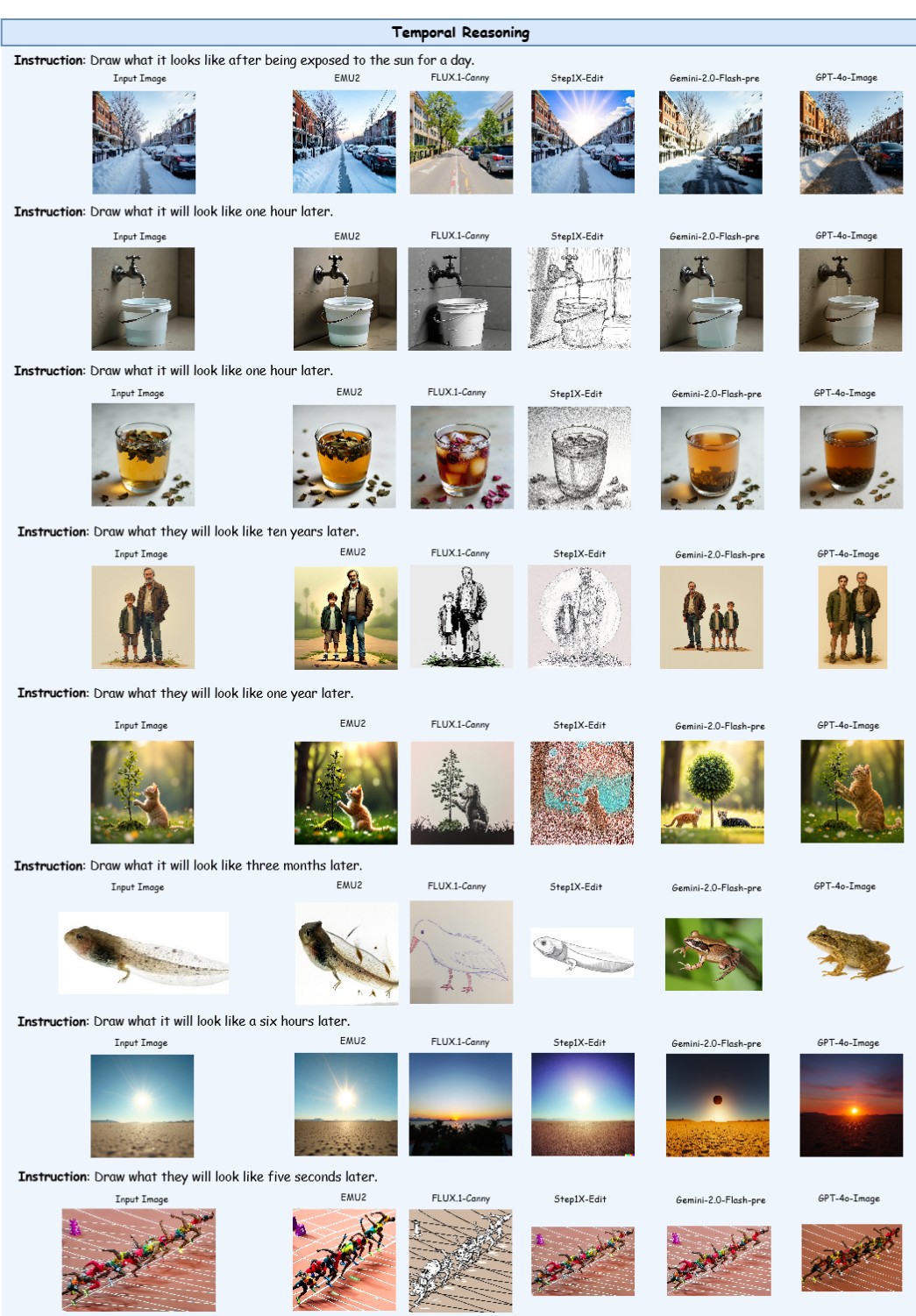

Figure 15: **Temporal Reasoning Outputs – Part 2.**

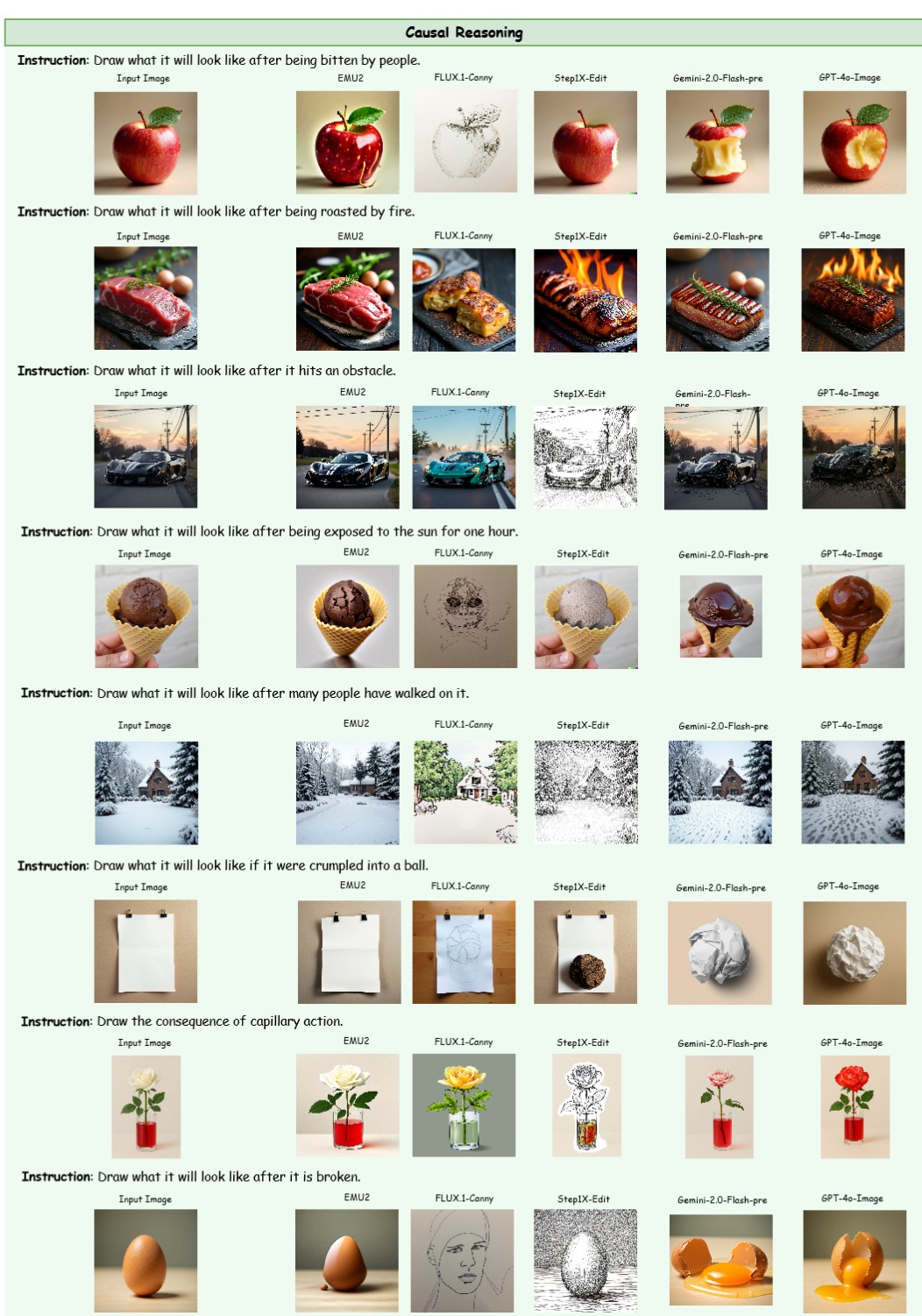

Figure 16: **Causal Reasoning Outputs – Part 1.**

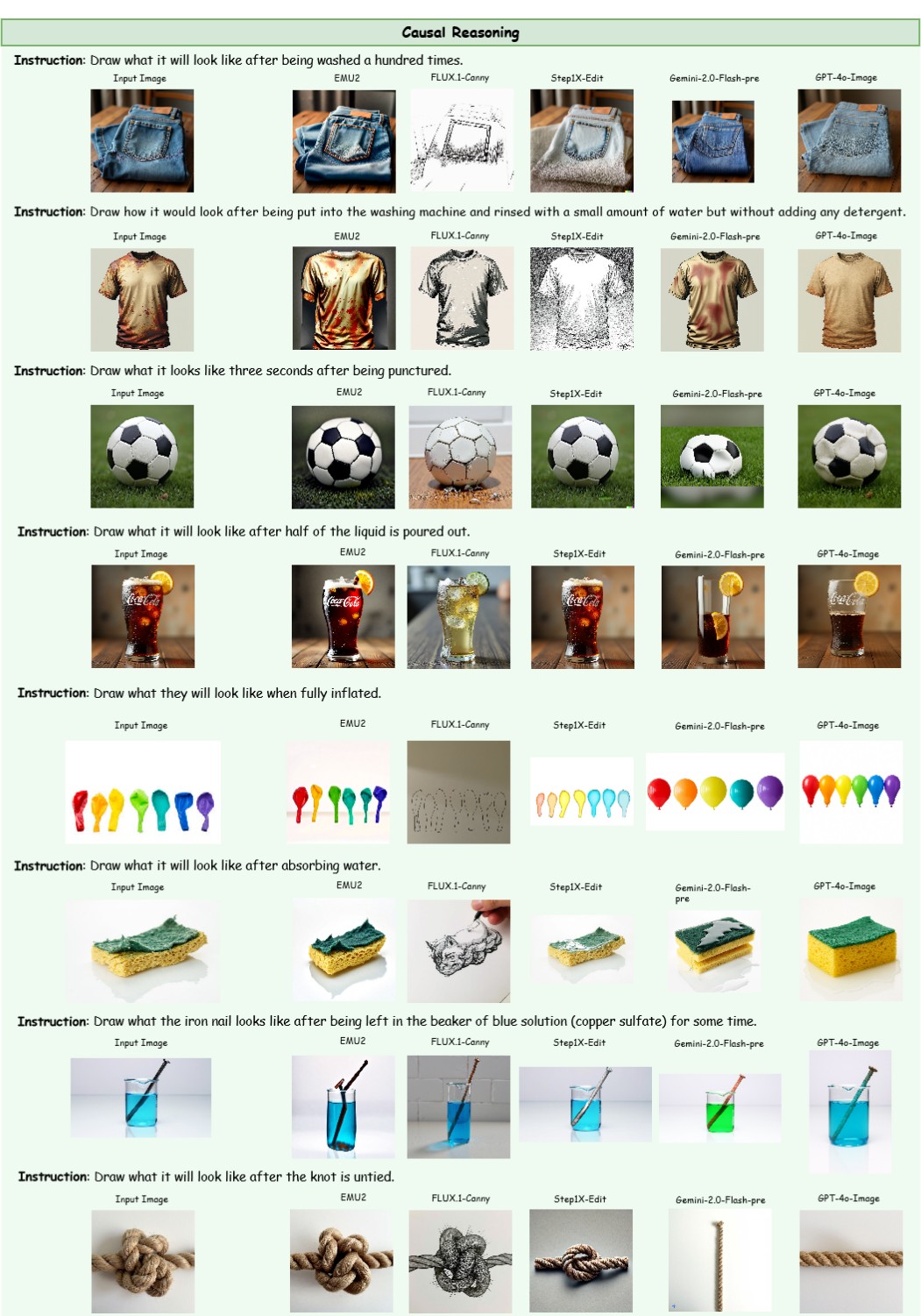

Figure 17: **Causal Reasoning Outputs – Part 2.**

## Prompt for Appearance Consistency on Temporal and Causal Reasoning

You are a highly skilled image evaluator. You will receive two images (an original image and a modified image) along with a specific modification instruction. The second image is known to have been altered based on this instruction, starting from the first image. Your task is to evaluate whether the two images maintain consistency in aspects not related to the given instruction.

Task Evaluate the consistency between the images according to the following scale (1 to 5):
- 5 (Perfect Consistency): Apart from changes explicitly required by the instruction, all other details (e.g., personal features, clothing, background, layout, colors, positions of objects) are completely identical between the two images.
- 4 (Minor Differences): Apart from changes explicitly required by the instruction, the second image is mostly consistent with the original image but contains a minor discrepancy (such as a missing minor personal feature, accessory, or tattoo).
- 3 (Noticeable Differences): Apart from changes explicitly required by the instruction, the second image has one significant difference from the original (such as a noticeable alteration in a person's appearance like hair or skin color, or a significant change in background environment).
- 2 (Significant Differences): Apart from changes explicitly required by the instruction, the second image has two or more significant differences or multiple noticeable inconsistencies (such as simultaneous changes in both personal appearance and background environment).
- 1 (Severe Differences): Apart from changes explicitly required by the instruction, nearly all key details (e.g., gender, major appearance features, background environment, or scene layout) significantly differ from the original image, clearly deviating from the original.
Example:
Original image: A blond, white-skinned man with a tattoo on his right shoulder, furniture in the background. Instruction: "Show him after gaining fifty pounds."
- Score 5: A heavier blond, white-skinned man, tattoo on right shoulder intact, identical furniture and layout.
- Score 4: A heavier blond, white-skinned man, missing the tattoo on his right shoulder, identical furniture and layout.
- Score 3: A heavier man with black hair instead of blond (change in hair color), or original blond man but with a grassy background instead of furniture.
- Score 2: A heavier man with black hair (hair color changed), and the background changed to grass.
- Score 1: A heavier black-haired woman, and background changed to grass.
Note: When assigning scores, only consider details unrelated to the instruction. Changes explicitly requested by the instruction should NOT be regarded as inconsistencies.
Input
Instruction: {instruct}
Output Format
Provide a detailed, step-by-step explanation of your scoring process. Conclude clearly with the final score, formatted as:
Final Score: 1-5

Figure 18: **Prompt for evaluating Appearance Consistency in Temporal Reasoning and Causal Reasoning.**

Prompt for Instruction Reasoning on Temporal and Causal Reasoning tasks.

You are an expert image evaluator. For each task, you will be provided with:
1. An instruction describing how an image should be modified.
2. A ground-truth textual description that represents the intended result of the modification.
3. An output image generated by an assistant.
Your task is to assess the output image based on the following evaluation dimension:
Evaluation Dimension: Alignment Between Image and Reference Description Assess how accurately the output image aligns with the visual content described in the reference description, considering the context of the instruction.
Scoring Criteria: - 5: The image completely matches the description, accurately reflecting every detail and degree.
- 4: The image mostly matches the description, with minor discrepancies.
- 3: The image partially matches the description but contains differences or lacks some details.
- 2: The image contains noticeable difference. Important details are missed or clearly inaccurate.
- 1: The image fails to follow the instruction and does not correspond to the description at all.
Example Instruction: Draw what it will look like after it is broken. Description: An egg is completely broken, with eggshell scattered around and egg white and yolk clearly spilling out.
- 5: Completely broken egg, clearly scattered eggshells, visible egg white and yolk spilling out.
- 4: Broken egg, eggshell present but not fully scattered, clearly visible egg white and yolk spilling out.
- 3: Broken egg with scattered eggshell, but egg white and yolk not spilled or still within eggshell.
- 2: Only scattered eggshell visible, without clear egg white or yolk.
- 1: Egg is intact, not broken.
Input Instruction instruct GroundTruth Description: reference
Output Format
Provide a detailed, step-by-step explanation of your scoring process. Conclude clearly with the final score, formatted as:
Final Score: X

Figure 19: **Prompt for evaluating Instruction Reasoning on Temporal and Causal Reasoning tasks.**

**Prompt for visual plausibility on Temporal and Causal Reasoning tasks.**

You are an expert image evaluator. For each task, you will be provided with an output image generated by an assistant.

Your task is to independently assess the image along the following dimension and assign an integer score from 1 to 5:

Evaluation Dimension: Realism and Generation Quality

Assess the overall visual realism and generation fidelity of the image. Consider the image's clarity, natural appearance, and compliance with physical plausibility and real-world constraints.

Scoring Guidelines:

- 5 The image is sharp, visually coherent, and all elements appear highly realistic and physically plausible.

- 4 The image is clear, with most elements appearing realistic; minor details may show slight unreality.

- 3 The image is mostly clear, but some significant elements appear unrealistic or physically implausible.

- 2 The image is noticeably blurry or contains major unrealistic components or visual distortions.

- 1 The image is extremely blurry, incoherent, or severely unrealistic; realism is nearly absent.

Output Format

After the evaluation, conclude clearly with the final score, formatted as:

Final Score: X

Figure 20: **Prompt for evaluating Visual Plausibility on Temporal and Causal Reasoning tasks.**

You are a precise and analytical image consistency evaluator.

You will be given: - Image A: the original image. - Image B: a modified version of Image A. - Instruction: a directive describing the intended modification to Image A to produce Image B. Your task is to evaluate how consistent Image B remains with Image A in all aspects *except* those explicitly changed by the instruction. You must ignore the instructed changes and only assess unintended differences.

Evaluation Scale (1 to 5):

- 5 Perfect Consistency All elements not related to the instruction are visually identical between Image A and Image B (e.g., style, background, object positions, colors, shapes). No unintended change is present.

- 4 Minor Difference One small unintended change is present (e.g., a slight color variation or minor object shape shift), but overall the image remains highly consistent.

- 3 Noticeable Difference One major or a few minor unintended changes are present (e.g., an object's shape, color, or background differs noticeably, or style has shifted slightly).

- 2 Significant Inconsistency Two or more significant differences unrelated to the instruction (e.g., changes in both object details and background or style), reducing overall fidelity.

- 1 Severe Inconsistency Major unintended changes dominate the image (e.g., altered visual style, scene layout, or appearance), clearly breaking consistency with Image A.

Note: - To receive a score of 5, the modified image must be visually identical to the original in every unaffected aspect—symbols, patterns, background, texture, color, category, layout, and style must all match exactly.

- If the background in the original is vague (e.g., plain white or composed of parts), and the background in Image B is also similar vague, you may disregard background consistency.

- If a blue diamond shape appears in the bottom-left corner of Image 2, ignore it; it is a watermark.

Example

Original image: "A silver-framed clock with a white face. Three hands (hour, minute, second) are disassembled and lie beside it." Instruction: "Assemble the clock to show 9:45."

Scoring Criteria: - Score 5: Frame, face, and hand shapes exactly as original.

- Score 4: One hand differs slightly in shape or thickness.

- Score 3: All hands identical, differing from original specs, or some other things(like text, furniture in the background) is added.

- Score 2: Frame color or face differs, and hand shapes are wrong.

- Score 1: Frame, face, and hand appearance all significantly altered, background is totally different.

Input Instruction: instruct

Output Format After evaluation, conclude with:

Final Score: 1-5

Figure 21: **Prompt for evaluating Appearance Consistency on Spatial Reasoning task.**

Prompt for evaluating Instruction Reasoning on Spatial Reasoning task.

You are an expert image evaluator. For each task, you will be provided with:
1. An instruction describing how an image should be modified. 2. A ground-truth textual description that represents the intended result of the modification. 3. An output image generated by an assistant.
Your task is to assess the output image based on the following evaluation dimension:
Evaluation Dimension: Alignment Between Image and Reference Description Assess how accurately the output image aligns with the visual content described in the reference description, considering the context of the instruction.
Scoring Criteria: - 5: The image completely matches the description, accurately reflecting every detail and degree.
- 4: The image mostly matches the description, with minor discrepancies.
- 3: The image partially matches the description but contains differences or lacks some details.
- 2: The image contains noticeable difference. Important details are missed or clearly inaccurate.
- 1: The image fails to follow the instruction and is entirely unrelated to the description.
Input Instruction instruct GroundTruth Description: reference
Output Format
Conclude clearly with the final score, formatted as:
Final Score: X

Figure 22: **Prompt for evaluating Instruction Reasoning on Spatial Reasoning task.**

Prompt for evaluating Visual Plausibility on Spatial Reasoning task.

You are a highly skilled image evaluator. Given an image, your task is to assess and determine its clarity and distortion, and then provide a score (an integer between 1 and 5) based on the following criteria:
Task Requirements:
Determine whether the image has blurriness, distortion, visual defects, or physical inaccuracies.
Assign an appropriate score to the image based on the above criteria, considering its overall quality and detail integrity.
Scoring Criteria:
- 5 points: The image is very clear, with complete details, and no noticeable distortion or blurriness. All elements conform to physical laws.
- 4 points: The image is clear, with only minor blurriness, and no noticeable distortion.
- 3 points: The image has areas with clarity issues, such as slight blurriness or distortion. Some elements are physically incorrect.
- 2 points: The image has noticeable blurriness or distortion, with significant detail loss, or lacks physical accuracy.
- 1 point: The image is severely blurry or distorted, making it difficult to recognize its content, with serious degradation in visual quality, almost unusable.
Output Format
Provide a clear conclusion with the final score, formatted as follows:
Final Score: 1-5
where X represents the score.

Figure 23: **Prompt for evaluating Visual Plausibility on Spatial Reasoning task.**

> **Prompt for evaluating Logical Reasoning Tasks with reference text answer.**
>
> You are a highly skilled image evaluator. Given an image with logical problem, you will receive:
> 1. Image 1: The original image. 2. Image 2: A generated image from an assistant model. 3. Problem Description 4. Reference Answer
> Your task is to determine whether Image 2 correctly match the reference answer. Evaluate Image 2 based on the following metrics, each scored as either 0 or 1:
> 1. Logical Correctness (0/1)
> - Assess whether the content of Image 2 logically matches the reference answer.
> - For example, given Image 1 is a teacher with "1+1=?" on the blackboard, and the problem is "Replace the question mark with the correct answer", if Image 2 replaces the question mark with "2", then the score is 1; other is 0.
> 2. Appearance Consistency (0/1)
> Determine whether the style, environment, arrangement of Image 2 are consistent with Image 1.
> - Consider factors such as color scheme, line/font style, background setting, etc. If Image 2's appearance fully aligns with Image 1, score 1; otherwise, score 0.
> - If the only difference is the actual problem solution (not the style or setting) or slightly lighter/darker color, still assign a score of 1.
> - If Image 2 is created by directly adding a pattern to Image 1, still assign a score of 1.
> - If in Image 1, the nodes and edges form an irregular quadrilateral with varying edge lengths and angles but form a square-like arrangement with equal edge lengths and right angles in Image 2, the score is 0.
> Inputs Problem Description: instruct Reference Answer: reference
> Output You should provide a step-by-step explanation of how you arrived at each score and conclude with the total scores for all three requirements in the format:
> Final Score: X,Y
> where X and Y are the scores for the two metrics (Logical Correctness and Appearance Consistency), respectively.

Figure 24: **Prompt for evaluating Logical Reasoning Tasks with reference text answer.**

> **Prompt for evaluating Logical Reasoning Tasks with reference image answer.**

You are a highly skilled image evaluator. Given a logical problem, you will receive:
1. Image 1: A reference ground-truth image that correctly solves the problem. 2. Image 2: A generated image from an assistant model.
Your task is to determine whether Image 2 correctly solves the problem, using Image 1 as the reference answer. Evaluate Image 2 based on the following metrics, each scored as either 0 or 1:
1. Logical Correctness (0/1)
Assess whether the content of Image 2 logically equal to Image 1.
Examples
- In a tic-tac-toe problem, if the positions of the marks in Image 2 are exactly the same as in Image 1, score 1; otherwise, score 0.
- If the problem is to , only if Image 2 is completely identical to Image 1(reference answer) in terms of shape, color, arrangement pattern, and pattern orientation, score 1; otherwise, score 0.
- If Image 1 only contains 1 gt answer but Image 2 contains several answers, score 0.
2. Appearance Consistency (0/1)
Determine whether the style and environment of Image 2 are consistent with Image 1.
- Consider factors such as color scheme, line style, background setting, etc. If Image 2's appearance fully aligns with Image 1, score 1; otherwise, score 0.
- If the only difference is the actual problem solution(such as Image 1 with red line as solution and Image 2 with blue line as solution) or slightly lighter/darker color, still assign a score of 1.
- If Image 2 is created by directly adding a pattern to Image 1, still assign a score of 1.
If a blue diamond shape appears in the bottom-left corner of Image 2, ignore it; it is a watermark.
Problem Description instruct
Output You should provide a step-by-step explanation of how you arrived at each score and conclude with the total scores for all three requirements in the format:
Final Score: X,Y
where X and Y are the scores for the two metrics (Logical Correctness and Appearance Consistency), respectively.

Figure 25: **Prompt for evaluating Logical Reasoning Tasks with reference image answer.**

