# OpenReview forum: "Envisioning Beyond the Pixels: Benchmarking Reasoning-Informed Visual Editing"
_NeurIPS.cc/2025/Datasets_and_Benchmarks_Track — NeurIPS 2025 Datasets and Benchmarks Track oral_

### Official Review · Reviewer_Euzw · 2025-06-25

**Rating:** 5
**Confidence:** 4

**Summary:**

This paper suggests RISEbench for evaluating reasoning-informed visual editing. Reasoning-informed visual editing means the visual editing task with a given prompt that contains the problem requiring the reasoning process to solve. They focus on 4 key reasoning categories, such as temporal, causal, spatial, and logical reasoning. To evaluate them properly, they propose three metrics named instruction reasoning (how well edited results follow instruction), appearance consistency (how well edited result preserves their original attributes), and visual plausibility (how well edited results are visually plausible). Evaluation is achieved by LLM to score the edited quality based on the given reference text or images. They evaluate 6 off-the-shelf generative models in terms of four reasoning categories and three metrics. They also validate that the LLM judgement is well-aligned with human annotators, especially identifying the critical failure of editing.

**Additional Feedback:**

- It would be great if authors report the source of every image. Currently, I only found their source from Appendix C., which ire reported somewhat shortly.

**Dataset Code Accessibility:**

Yes

**Dataset Code Comments:**

All of the images are available at the provided GitHub page with labels in a JSON file.

They also provide evaluation codes with GPT-4.1.

**Ethical Considerations:**

No, there are no or only very minor ethics concerns

**Final Justification:**

Authors clarify the data source of the proposed dataset.

Also they mentioned that they will resolve the lack of the size of dataset.

Thus, my concerns are resolved and keep my original rating.

**Limitations Weaknesses:**

- There are no notable weaknesses or limitations

- One minor limitation is the number of images in the benchmark (360 images), but this is acceptable as it is the first benchmark for reasoning-informed visual editing benchmark.

**Strengths Contributions:**

- The task of reasoning-informed visual editing is an interesting and significant problem.

- Reasoning categories and evaluation metrics seem to make sense.

- Checking the correlation between LLM judges and human evaluators effectively verifies the validity of the proposed benchmark.

---

> ### Author Rebuttal · Authors · 2025-07-31
>
> Dear Reviewer Euzw:
>
> Thank you for your constructive and supportive comments. We're glad that you found the task formulation, evaluation metrics, and validation strategy meaningful. We hope our responses below will further affirm your positive impression.
>
> ### Limitations Weaknesses
>
> **W1: One minor limitation is the number of images in the benchmark (360 images), but this is acceptable as it is the first benchmark for reasoning-informed visual editing benchmark.**
>
> A1: We sincerely thank the reviewer for the thoughtful comment regarding the scale and scope of RISEBench.
>
> While RISEBench currently comprises 360 meticulously curated examples, our design philosophy for this initial version prioritized **quality and diversity over sheer quantity**. Each instance was carefully selected to ensure high quality, a broad spectrum of diverse instructions, and rich, complex editing scenarios. This deliberate focus allows our benchmark to effectively evaluate a wide range of model capabilities within a manageable dataset size.
>
> We will seriously consider increasing the number of examples per task type, enriching sub-task categories and incorporating more complex settings such as multi-turn and interactive editing, video, and 3D tasks in future versions of the benchmark, as also suggested by Reviewers UK1u and MJQz.
>
> Thank you once again for your valuable input.
>
> **W2: It would be great if authors report the source of every image. Currently, I only found their source from Appendix C., which ire reported somewhat shortly.**
>
> A2: We are very grateful to the reviewer for their pertinent concern regarding the source of each sample within RISEBench. This is a crucial point for transparency and reproducibility.
>
> Addressing your valuable comments, we have meticulously analyzed and clarified the approximate proportions of data sources for each task category, as detailed below:
>
> - **Temporal Reasoning**: Around 49% of examples are generated by advanced image generation models, with the remaining 51% collected from diverse sources on the Internet.
>
> - **Causal Reasoning**: Around 41% of samples originate from image generation models, while approximately 59% are collected from the Internet.
>
> - **Spatial Reasoning**: Around 56% of samples are synthetically rendered from controlled 3D environments (e.g., using Blender), and the remaining 44% are collected from the Internet.
>
> - **Logical Reasoning**: Around 53% of the instances are derived or adapted from existing datasets and benchmarks, with the remaining 47% sourced from the Internet.
>
> We will promptly update **Appendix C** in the revised manuscript to reflect this detailed approximate distribution and to enhance the clarity of source attributions for each task type.
>
> Thank you once again for your insightful suggestions, which have helped us improve the clarity of our dataset documentation.

---

> > ### Comment · Reviewer_Euzw · 2025-08-05
> >
> > Thanks for the author's rebuttal, and I hope the number of examples will be extended further.

---

> > > ### Author Response · Authors · 2025-08-05
> > >
> > > Dear Reviewer Euzw,
> > >
> > > We sincerely appreciate your encouraging comments and follow-up suggestion. Expanding the number of examples is indeed one of our key next steps, and we look forward to enriching the benchmark in future iterations. Thank you again for your thoughtful and constructive feedback throughout the review process.
> > >
> > > Authors

---

### Official Review · Reviewer_MJQz · 2025-06-30

**Rating:** 5
**Confidence:** 4

**Summary:**

The paper introduces RISEBench, the first benchmark dedicated to Reasoning-Informed Visual Editing (RISE). RISEBench focuses on assessing large multi-modality models (LMMs) on their ability to understand and edit images based on four types of reasoning: 1. Temporal Reasoning, 2. Causal Reasoning, 3. Spatial Reasoning, 3. Logical Reasoning.

The benchmark evaluates the models based on three dimensions:

1. Instruction Reasoning: How well the model understands and follows the editing instructions.

2. Appearance Consistency: Whether the model preserves the non-edited parts of the image.

3. Visual Plausibility: How realistic and coherent the final image appears.

The paper tests several models, including both open-source (e.g., GPT-4o-Image, Gemini-2.0-Flash) and proprietary models, showing that while models like GPT-4o-Image outperform others, they still struggle with logical reasoning tasks. The evaluation pipeline uses LMM-as-a-judge, which provides scalable, automated assessments and aligns well with human evaluations.

**Additional Feedback:**

n/a

**Dataset Code Accessibility:**

Yes

**Dataset Code Comments:**

n/a

**Ethical Considerations:**

No, there are no or only very minor ethics concerns

**Final Justification:**

Thanks for the authors' rebuttal. I think these findings are inspiring and logical. I maintain my score as Accept.

**Limitations Weaknesses:**

1. While RISEBench excels at reasoning-informed editing, it is focused on relatively simple edits involving static images. The lack of multi-turn or interactive editing tasks limits its applicability to real-world scenarios, where images often undergo a series of edits or adjustments.

2. Logical reasoning is presented as a major challenge for current models, but the paper lacks an in-depth analysis of the failure modes. The limited performance on logical tasks suggests the need for specialized models or algorithms focused on logic-based reasoning in visual contexts.

**Strengths Contributions:**

1. The emphasis on reasoning-informed editing (RISE) is a timely contribution. The ability to perform tasks such as adjusting images based on temporal evolution or logical puzzles is a major step forward in the field of generative image editing and is likely to influence future model development.

2. The RISEBench is a valuable and novel contribution, addressing a critical gap in the evaluation of reasoning in image editing. By targeting temporal, causal, spatial, and logical reasoning, it provides a more comprehensive view of model capabilities than traditional text-to-image or image-editing benchmarks.

3. The inclusion of three evaluation dimensions (Instruction Reasoning, Appearance Consistency, Visual Plausibility) is well thought-out and provides a thorough assessment of model performance across different aspects of image editing. The LMM-as-a-judge approach adds scalability and consistency to the evaluation process.

---

> ### Author Rebuttal · Authors · 2025-07-31
>
> Dear Reviewer MJQz:
>
> Thank you for your detailed and encouraging feedback. We're pleased that you recognized the relevance of reasoning-informed editing and the value of our benchmark design. We hope our responses below further substantiate your positive assessment.
>
> ### Limitations Weaknesses
>
> **W1: While RISEBench excels at reasoning-informed editing, it is focused on relatively simple edits involving static images. The lack of multi-turn or interactive editing tasks limits its applicability to real-world scenarios, where images often undergo a series of edits or adjustments.**
>
> A1: We fully agree with the reviewer that multi-turn and interactive editing paradigms are indeed crucial for better reflecting real-world image editing scenarios.
>
> While the primary focus of our current work is on single-turn editing, a deliberate choice made to ensure the quality, clarity, and foundational robustness of our initial benchmark, we deeply appreciate the importance of extending RISEBench in the direction of more complex, sequential interactions. We will seriously consider incorporating multi-turn or interactive editing tasks in future versions of the benchmark. This aligns with our broader vision for developing comprehensive and ecologically valid evaluation tools for advanced image editing models.
>
> Thank you for this valuable and forward-looking suggestion.
>
> **W2: Logical reasoning is presented as a major challenge for current models, but the paper lacks an in-depth analysis of the failure modes. The limited performance on logical tasks suggests the need for specialized models or algorithms focused on logic-based reasoning in visual contexts.**
>
> A2: We appreciate the reviewer's insightful question regarding the performance of current generative models on reasoning tasks.
>
> Indeed, as our experimental results unequivocally indicate, current state-of-the-art generative models exhibit significant limitations in fundamental logic-based reasoning capabilities for visual editing. Even the strongest evaluated model, GPT-4o-Image, achieves a remarkably low accuracy of merely 10.6% on logical reasoning tasks. This starkly highlights a substantial gap in performance compared to other reasoning categories within our benchmark.
>
> Through a preliminary qualitative analysis, we have identified several recurring failure modes that contribute to these limitations:
>
> - **Instruction Misinterpretation**: Models frequently fail to correctly parse and execute structured or rule-based tasks, such as Sudoku-like puzzles, pattern assembly, or path planning.
>
> - **Failure in Multi-step Reasoning**: Tasks that necessitate sequential inference and intermediate transformations (e.g., complex puzzle solving or symbolic transformation) prove particularly challenging. This is likely attributable to the models' limited ability to track and maintain consistency across intermediate states during multi-step logical derivations.
>
> - **Visual Inconsistency despite Logical Correctness**: In some instances, while the generated output might be conceptually or logically correct in response to the instruction, it is ultimately rejected due to significant deviations in layout, style, or color when compared to the original input image.
>
> We firmly believe that our RISEBench effectively surfaces these critical limitations, serving as a valuable diagnostic tool for the community. As more capable and specialized models emerge—especially those explicitly designed or fine-tuned for robust logic-based visual reasoning—we are committed to incorporating and rigorously evaluating them within the framework of RISEBench to track progress in this crucial area.
>
> Thank you once again for prompting this important discussion on model capabilities.

---

> > ### Comment · Reviewer_MJQz · 2025-08-05
> > **response to the rebuttal**
> >
> > Thanks to the authors' clarification and comments, which are inspiring and logical. I don't have other questions.

---

> > > ### Author Response · Authors · 2025-08-05
> > >
> > > Dear Reviewer MJQz,
> > >
> > > We sincerely thank you for your thoughtful feedback and encouraging words. We're glad that our clarifications resonated with you, and we truly appreciate your valuable time and support throughout the review process.
> > >
> > > Authors

---

### Official Review · Reviewer_KS6A · 2025-07-01

**Rating:** 5
**Confidence:** 4

**Summary:**

This article presents RISEBench, a benchmark designed to evaluate Reasoning-Informed Visual Editing (RISE) within Large Multi-modality Models (LMMs). RISEBench focuses on four critical reasoning categories: Temporal, Causal, Spatial, and Logical Reasoning, for which high-quality test cases have been curated. A robust evaluation framework is then proposed to assess Instruction Reasoning, Appearance Consistency, and Visual Plausibility, utilizing both human judges and the LMM-as-a-judge approach. Experimental results demonstrate that RISEBench is a reliable benchmark for evaluating reasoning-informed visual editing. The findings highlight the limitations of current editing models and provide valuable insights, indicating potential future directions for the development of reasoning-aware visual editing.

**Dataset Code Accessibility:**

Yes

**Ethical Considerations:**

No, there are no or only very minor ethics concerns

**Final Justification:**

The response provides a clear and technically rigorous rationale that directly addresses concerns about compressed score variance, showing that the observed behavior is consistent with expectations. Moreover, the justification of the LMM-as-Judge framework demonstrates its reliability and evaluative robustness, which alleviates potential doubts. Overall, the clarity and thoroughness of the response substantiate the assigned score.

**Limitations Weaknesses:**

Concerns and Questions:

1. The quantitative results presented in Table 1 exhibit sparsity, with scores predominantly clustered in the lower range. This raises the question of whether the benchmark's high difficulty level limits its ability to effectively discriminate performance differences among the evaluated models.

2. Within the evaluation framework of **Dimension 2: Appearance Consistency**, Large Multimodal Models (LMMs) are employed for automated assessment. My concern is whether LMMs can effectively capture the identity and details of objects, as this may impact the accuracy of the evaluation.

**Strengths Contributions:**

Pros:

1. RISEBench offers a comprehensive and hierarchically organized categorization of reasoning tasks, with well-balanced distributions across different dimensions. This structured design establishes RISE as a robust and versatile benchmark for evaluating Reasoning-Informed Image Editing.

2. The design of the experiment is reasonable and sufficient, and the analysis of the experimental results also brings insights for future visual editing research.

3. This paper is well-structured and easy to follow. The description of the construction and evaluation methods of RISEBench is clear and straightforward.

---

> ### Author Rebuttal · Authors · 2025-07-31
>
> Dear Reviewer KS6A:
>
> We sincerely appreciate your encouraging comments. It's great to know that the structure and clarity of our work resonated with you. We hope our replies below further support your positive assessment.
>
> ### Limitations Weaknesses:
>
> **W1: The quantitative results presented in Table 1 exhibit sparsity, with scores predominantly clustered in the lower range. This raises the question of whether the benchmark's high difficulty level limits its ability to effectively discriminate performance differences among the evaluated models.**
>
> A1: We are very grateful to the reviewer for their insightful question regarding our strict success criterion.
>
> We acknowledge that a stringent success criterion, by design, may indeed compress score variance. However, this metric was deliberately chosen to reflect the high bar required in real-world image editing applications. In such practical scenarios, a failure in even one critical dimension can render an output unusable, making overall generation quality paramount. This design choice directly aligns with our primary goal of ensuring the practical utility and robustness of the generated outputs.
>
> To provide a better understanding of model performance and facilitate comparative analysis, we have already reported the average scores for each individual dimension (as shown in Figure 5 of the original paper). These per-dimension scores effectively reveal subtle differences between models that might otherwise be obscured by the strict aggregate metric. For instance, while both HiDream-Edit and EMU2 might achieve 0% under the strict combined success criterion, their individual dimensional scores tell a much more detailed story.
>
> Specifically, HiDream-Edit demonstrates superior performance in Instruction Reasoning (30.3 vs. 22.6), suggesting a stronger capability in interpreting complex instructions. Conversely, EMU2 significantly outperforms HiDream-Edit in Appearance Consistency (38.2 vs. 12.6) and also shows stronger results in Visual Plausibility (78.3 vs. 74.9). This detailed breakdown clearly indicates that, despite a similar overall strict success score, HiDream-Edit possesses better reasoning capabilities, while EMU2 excels in maintaining visual coherence and plausibility.
>
> We will consider incorporating softer metrics in future work that could further enhance our model analysis, and we will certainly consider this valuable suggestion for subsequent research.
>
> **W2: Within the evaluation framework of Dimension 2: Appearance Consistency, Large Multimodal Models (LMMs) are employed for automated assessment. My concern is whether LMMs can effectively capture the identity and details of objects, as this may impact the accuracy of the evaluation.**
>
> A2: We sincerely appreciate the reviewer's insightful comments regarding the correctness and nuanced capabilities of LMM-as-Judge frameworks.
>
> Indeed, we acknowledge that while current LMMs used as judges may exhibit certain limitations in capturing highly granular, pixel-level details, their overall evaluation accuracy, particularly in broader aspects of appearance consistency, remains demonstrably reliable. As presented in **Table 2**, the MAE between LMM and human scores for Appearance Consistency is merely 0.7. This low MAE value strongly indicates a high degree of alignment with human judgment on the broader dimension of visual consistency.
>
> Interestingly, LMMs can sometimes exhibit a superior ability to detect subtle, unintended changes that human experts might inadvertently overlook. For example, in the case of *temporal_reasoning_16*, GPT-4.1 accurately identified nuanced inconsistencies in personal attributes, clothing details, background elements, and overall layout between two images, consequently assigning a consistency score of 2. In contrast, human raters provided a noticeably higher average score of 3.5. This observation suggests that LMMs may adhere more strictly and objectively to the predefined evaluation metric, whereas human raters might occasionally introduce subjective leniency or overlook minute deviations.
>
> However, we fully agree with the reviewer that LMMs sometimes struggle with truly fine-grained, pixel-level appearance details, particularly concerning very small symbols or textual elements. A notable instance is *temporal_reasoning_61*, where the LMM failed to detect subtle differences in keycap characters, leading to slight hallucinations in its assessment. While this phenomenon exists, it is relatively infrequent in our dataset and, in most contexts, the overall reliability of LMM-based evaluation holds. To further elucidate these specific strengths and limitations of LMM-based assessments in evaluating Appearance Consistency, we will include additional qualitative examples and detailed discussions in the revised paper.
>
> We will  further investigate this aspect as part of our future work to ensure even more precise and reliable automated evaluations. Thank you once again for your valuable feedback, which has prompted us to provide a more nuanced understanding of our evaluation methodology.

---

> > ### Comment · Reviewer_KS6A · 2025-08-07
> >
> > Thank you for the detailed and thoughtful response.
> >
> > The explanation is sound and effectively addresses my concerns regarding the compressed score variance and the capabilities of the LMM-as-Judge framework.

---

> > > ### Author Response · Authors · 2025-08-07
> > >
> > > Dear Reviewer KS6A,
> > >
> > > Thank you very much for your thoughtful feedback and kind recognition. We’re glad our clarifications addressed your concerns effectively. Your insights have been very helpful in improving the quality of our work.
> > >
> > > Authors

---

### Official Review · Reviewer_UK1u · 2025-07-03

**Rating:** 5
**Confidence:** 4

**Summary:**

This paper presents RISEBench, a new benchmark for evaluating visual editing models on tasks that require complex reasoning. It covers four types of reasoning—temporal, causal, spatial, and logical—with 360 annotated examples. The authors introduce a three-part evaluation protocol and use a GPT-4-based “LMM-as-a-Judge” pipeline, validated against human ratings. Experiments on eight models show that current methods struggle with reasoning-based editing, with even the best model achieving less than 30% accuracy.

**Additional Feedback:**

My main concerns lie in the first two limitations above. The current strict success criterion (i.e., requiring 5/5 on all aspects) may mask meaningful differences between models and compress performance spread. Introducing softer evaluation metrics could improve the reliability and robustness of the assessment.

Second, relying solely on GPT-4.1 as the evaluator raises fairness concerns. Incorporating an alternative evaluator (e.g., Gemini-Pro or an open-source model) would help verify consistency and mitigate potential bias.

Overall, the benchmark targets an important and underexplored area. It represents a promising direction and provides a solid foundation for evaluating reasoning-aware visual editing.

**Dataset Code Accessibility:**

Yes

**Dataset Code Comments:**

Both the dataset and evaluation code are publicly accessible, with clear documentation and usage instructions that support reproducibility.

**Ethical Comments:**

No major concerns.
- Images are either synthetic, rendered, or licensed (Appx C); privacy/consent issues minimal.

- The study focuses on evaluation and does not involve human subjects or sensitive content; ethical risks are minimal.

**Ethical Considerations:**

No, there are no or only very minor ethics concerns

**Final Justification:**

The explanation of the evaluation metric and the clarification of how per-aspect scores reflect model differences address my concern about the compressed performance variance. I also appreciate the additional experiments comparing different LMM evaluators, which improve the robustness of the evaluation.

I hope the authors can incorporate these clarifications—particularly regarding the evaluation criterion and the comparison involving diverse or open-sourced evaluators—into the revised version to support deeper insights and facilitate broader and more accessible evaluation.

**Limitations Weaknesses:**

**Strict Success Criterion Compresses Model Differences**：
The benchmark adopts an all-or-nothing accuracy metric: a sample is only counted as correct if it scores a perfect 5/5 on all three aspects. This rigid threshold compresses score variance and fails to reflect partial successes or relative strengths across models. As a result, models with qualitatively different capabilities may appear equally poor. For instance, Table 1 shows that both HiDream-Edit and FLUX.1-Canny receive 0% overall, despite possibly differing in reasoning ability. Introducing softer metrics (e.g.,  threshold or weighted scoring) would better capture such distinctions.

**Single Evaluator Risk in LMM-as-a-Judge**：
The evaluation relies solely on GPT-4.1, raising potential fairness and bias concerns. While the paper includes human alignment studies, incorporating additional judges (e.g., Claude, Gemini, or Qwen-VL) and comparing their agreement with human ratings would improve robustness and offer insights into how different LMMs align with human evaluation. Open-source evaluators could also enhance transparency and reproducibility.

**Limited Dataset Scale**
RISEBench includes only 360 examples, which may limit statistical power and model generalization across diverse editing instructions; future versions could modestly expand to formats like video, 3D, or multi-turn editing to cover broader reasoning scenarios.

**Strengths Contributions:**

**RISEBench introduces a novel benchmark** for reasoning-informed visual editing, a gap unaddressed by prior generation or understanding datasets. Its focus on structured reasoning types—temporal, causal, spatial, logical—is timely and valuable.

**The evaluation dimensions are well-chosen**, with three core dimensions—instruction following, appearance consistency, and visual plausibility—capturing essential aspects of real-world editing quality.

**The LMM-as-a-Judge pipeline is practical and validated**, achieving strong consistency with human judgments (MAE ≤ 0.7), and enabling scalable, interpretable assessments without the overhead of manual scoring.

**Results are thoroughly analyzed**, with detailed breakdowns across the four reasoning types (temporal, causal, spatial, logical) and three evaluation dimensions. This granularity helps pinpoint specific failure modes and strengths of different model families.

---

> ### Author Rebuttal · Authors · 2025-07-31
>
> Dear Reviewer UK1u:
>
> Thank you for your thoughtful and positive feedback on our work. We are pleased that you found our benchmark, evaluation design, and analysis valuable. We hope that our responses below will address your concerns and reinforce your positive evaluation.
>
> ### Limitations Weaknesses
>
> **W1: Strict Success Criterion Compresses Model Differences： The benchmark adopts an all-or-nothing accuracy metric: a sample is only counted as correct if it scores a perfect 5/5 on all three aspects. This rigid threshold compresses score variance and fails to reflect partial successes or relative strengths across models. As a result, models with qualitatively different capabilities may appear equally poor. For instance, Table 1 shows that both HiDream-Edit and FLUX.1-Canny receive 0% overall, despite possibly differing in reasoning ability. Introducing softer metrics (e.g., threshold or weighted scoring) would better capture such distinctions.**
>
> A1: We are very grateful to the reviewer for their insightful question regarding our strict success criterion.
>
> We acknowledge that a stringent success criterion, by design, may indeed compress score variance. However, this metric was deliberately chosen to reflect the high bar required in real-world image editing applications. In such practical scenarios, a failure in even one critical dimension can render an output unusable, making overall generation quality paramount. This design choice directly aligns with our primary goal of ensuring the practical utility and robustness of the generated outputs.
>
> To provide a better understanding of model performance and facilitate comparative analysis, we have already reported the average scores for each individual dimension (as shown in Figure 5 of the original paper). These per-dimension scores effectively reveal subtle differences between models that might otherwise be obscured by the strict aggregate metric. For instance, while both HiDream-Edit and EMU2 might achieve 0% under the strict combined success criterion, their individual dimensional scores tell a much more detailed story.
>
> Specifically, HiDream-Edit demonstrates superior performance in Instruction Reasoning (30.3 vs. 22.6), suggesting a stronger capability in interpreting complex instructions. Conversely, EMU2 significantly outperforms HiDream-Edit in Appearance Consistency (38.2 vs. 12.6) and also shows stronger results in Visual Plausibility (78.3 vs. 74.9). This detailed breakdown clearly indicates that, despite a similar overall strict success score, HiDream-Edit possesses better reasoning capabilities, while EMU2 excels in maintaining visual coherence and plausibility.
>
> We will consider incorporating softer metrics in future work that could further enhance our model analysis, and we will certainly consider this valuable suggestion for subsequent research.
>
> **W2: Single Evaluator Risk in LMM-as-a-Judge： The evaluation relies solely on GPT-4.1, raising potential fairness and bias concerns. While the paper includes human alignment studies, incorporating additional judges (e.g., Claude, Gemini, or Qwen-VL) and comparing their agreement with human ratings would improve robustness and offer insights into how different LMMs align with human evaluation. Open-source evaluators could also enhance transparency and reproducibility.**
>
> A2: We are very grateful to the reviewer for their insightful question regarding the inclusion of additional models as evaluators.
>
> To comprehensively address this concern and further assess the robustness and alignment of various LLM-based evaluators with human judgments, we have conducted additional experiments incorporating two strong alternative multi-modal models: **Qwen2.5-VL-72B** and **Gemini-2.0-Flash**.
>
> Our updated analysis consistently demonstrates that **GPT-4.1** exhibits the closest alignment with human ratings. It achieves the lowest overall MAE across the Reasoning (0.5), Consistency (0.7), and Plausibility (0.4) dimensions (as shown in Table 2 of the original paper). In contrast, **Qwen2.5-VL-72B** and **Gemini-2.0-Flash** yield notably higher MAEs, particularly in the Reasoning and Consistency dimensions (e.g., **Qwen2.5-VL-72B**: MAE = 0.8/1.2; **Gemini-2.0-Flash**: MAE = 1.0/1.0).
>
> Table: Qwen2.5-VL-72B
>
> | Score | Prop (R/C/P)       | Human Mean (R/C/P)   | Human Std. (R/C/P)    | Mean Error (R/C/P)   | MAE (R/C/P)         |
> |-------|--------------------|----------------------|------------------------|----------------------|---------------------|
> | 1     | 25% / 0% / 0%      | 1.2 / – / –          | 0.6 / – / –            | 0.2 / – / –          | 0.2 / – / –         |
> | 2     | 8% / 5% / 0%       | 1.7 / 3.2 / –        | 0.8 / 0.5 / –          | -0.3 / 1.2 / –       | 0.8 / 1.2 / –       |
> | 3     | 17% / 12% / 0%     | 4.3 / 3.8 / –        | 0.8 / 0.7 / –          | 1.3 / 0.8 / –        | 1.3 / 0.8 / –       |
> | 4     | 17% / 8% / 30%     | 4.1 / 3.9 / 4.4      | 1.3 / 0.6 / 0.6        | 0.1 / -0.1 / 0.4     | 1.1 / 0.5 / 0.7     |
> | 5     | 33% / 75% / 70%    | 4.7 / 4.5 / 4.8      | 0.4 / 0.5 / 0.3        | -0.3 / -0.5 / -0.2   | 0.3 / 0.5 / 0.2     |
> | **Overall** | –            | –                    | –                      | –                    | **0.8 / 1.2 / 0.4** |
>
> Table: Gemini-2.0-Flash
>
> | Score | Prop (R/C/P)       | Human Mean (R/C/P)   | Human Std. (R/C/P)    | Mean Error (R/C/P)   | MAE (R/C/P)         |
> |-------|--------------------|----------------------|------------------------|----------------------|---------------------|
> | 1     | 25% / 5% / 0%      | 1.3 / 2.7 / –        | 0.7 / 0.5 / –          | 0.3 / 1.7 / –        | 0.3 / 1.7 / –       |
> | 2     | 4% / 5% / 0%       | 4.2 / 3.8 / –        | 0.9 / 0.5 / –          | 2.2 / 1.8 / –        | 2.2 / 1.8 / –       |
> | 3     | 8% / 4% / 1%       | 3.7 / 4.3 / 4.7      | 1.3 / 0.8 / 0.0        | 0.7 / 1.3 / 1.7      | 1.3 / 1.3 / 1.7     |
> | 4     | 12% / 2% / 11%     | 3.9 / 3.8 / 4.6      | 1.2 / 0.1 / 0.2        | -0.1 / -0.2 / 0.6    | 1.0 / 0.2 / 0.6     |
> | 5     | 51% / 84% / 88%    | 4.7 / 4.5 / 4.7      | 0.5 / 0.6 / 0.5        | -0.3 / -0.5 / -0.3   | 0.3 / 0.5 / 0.3     |
> | **Overall** | –            | –                    | –                      | –                    | **1.0 / 1.0 / 0.4** |
>
> These results confirm that while the inclusion of additional evaluators provides valuable cross-validation and insights into the landscape of LLM-based evaluation, **GPT-4.1** remains the most reliable automated judge among those tested, especially in terms of agreement with our expert human annotations. We will add the new tables to the **Appendix** to reflect these findings.
>
> Thank you once again for this valuable suggestion, which has significantly strengthened our evaluation methodology.
>
> **W3: Limited Dataset Scale RISEBench includes only 360 examples, which may limit statistical power and model generalization across diverse editing instructions; future versions could modestly expand to formats like video, 3D, or multi-turn editing to cover broader reasoning scenarios.**
>
> A3: We appreciate the reviewer's insightful question regarding the scale of our RISEBench.
>
> While RISEBench currently comprises 360 examples, the design principle behind its scale is rooted in a focus on **quality** and **diversity** rather than mere quantity. Each instance within the benchmark has been meticulously curated to ensure high quality and to encompass a broad spectrum of instruction types and scene contexts. This deliberate focus on diversity across question types is crucial for effectively assessing the general reasoning capabilities of image editing models. Therefore, our design choice prioritizes the breadth of covered capabilities over expanding the sheer volume of examples for each specific question type.
>
> Furthermore, our experimental results underscore the benchmark's effectiveness even at its current scale. The observed substantial performance gaps among various models (as detailed in our experiments) clearly indicate that RISEBench is sufficiently challenging and discriminative for distinguishing different model capabilities.
>
> We fully concur with the reviewer that expanding the benchmark to incorporate more complex formats, such as video, 3D, or multi-turn editing, represents a highly promising and necessary direction for future research. We are keen to explore such extensions in our subsequent work to further advance the field of image editing evaluation.
>
> Thank you once again for raising this important point.

---

> > ### Comment · Reviewer_UK1u · 2025-08-05
> > **Feedback to authors**
> >
> > Thank you for addressing my concerns.
> >
> > The explanation of the evaluation metric and the clarification of how per-aspect scores reflect model differences address my concern about the compressed performance variance. I also appreciate the additional experiments comparing different LMM evaluators, which improve the robustness of the evaluation.
> >
> > I will raise my score accordingly. I hope the authors can incorporate these clarifications—particularly regarding the evaluation criterion and the comparison involving diverse or open-sourced evaluators—into the revised version to support deeper insights and facilitate broader and more accessible evaluation.

---

> > > ### Author Response · Authors · 2025-08-06
> > >
> > > Dear Reviewer UK1u,
> > >
> > > Thank you very much for your thoughtful feedback and kind appreciation.
> > >
> > > We will incorporate the clarifications on the evaluation criterion and the additional evaluator comparisons into the revised version, as suggested.
> > >
> > > We sincerely appreciate your support in improving our work.
> > >
> > > Authors

---

### Decision · Program_Chairs · 2025-09-18

**Decision:**

Accept (oral)

**Comment:**

This paper introduces RISEBench, a novel benchmark for evaluating Reasoning-Informed Visual Editing (RISE) capabilities of Large Multi-modality Models (LMMs). The benchmark categorizes reasoning into temporal, causal, spatial, and logical, and proposes an evaluation framework assessing instruction reasoning, appearance consistency, and visual plausibility using both human and LMM-as-a-judge approaches. The findings highlight significant limitations in current models, with even the best performing model achieving low accuracy.

Four reviews were received, all recommending 5 (Accept).

A key strength of the paper is the introduction of a timely and valuable benchmark addressing a gap in evaluating reasoning in image editing, with a well-structured evaluation that aligns with human judgment. Concerns were raised regarding the strict success criterion potentially masking model differences, the reliance on a single LMM evaluator, and the dataset's scale. The authors addressed these by explaining that the strict metric aligns with real-world application needs and that per-dimension scores reveal finer differences. They also performed additional experiments with other LMM evaluators, confirming GPT-4.1's alignment with human ratings, and justified the dataset size by emphasizing quality and diversity, while acknowledging plans for future expansion.

Given the consistent positive ratings and the authors' thorough addressing of all concerns, the paper is recommended for acceptance.